# Sharp Analysis of Stochastic Optimization under Global Kurdyka-Łojasiewicz Inequality

**Ilyas Fatkhullin**[*]
ETH AI Center & ETH Zurich

**Jalal Etesami**[*]
EPFL[†]

**Niao He**
ETH Zurich

**Negar Kiyavash**
EPFL[†]

## Abstract

We study the complexity of finding the global solution to stochastic nonconvex optimization when the objective function satisfies global Kurdyka-Łojasiewicz (KŁ) inequality and the queries from stochastic gradient oracles satisfy mild expected smoothness assumption. We first introduce a general framework to analyze Stochastic Gradient Descent (SGD) and its associated nonlinear dynamics under the setting. As a byproduct of our analysis, we obtain a sample complexity of $\mathcal{O}(\epsilon^{-(4-\alpha)/\alpha})$ for SGD when the objective satisfies the so called $\alpha$-PŁ condition, where $\alpha$ is the degree of gradient domination. Furthermore, we show that a modified SGD with variance reduction and restarting (PAGER) achieves an improved sample complexity of $\mathcal{O}(\epsilon^{-2/\alpha})$ when the objective satisfies the average smoothness assumption. This leads to the first optimal algorithm for the important case of $\alpha = 1$ which appears in applications such as policy optimization in reinforcement learning.

## 1  Introduction

Nonconvex optimization problems are ubiquitous in machine learning domains such as training deep neural networks [22] or policy optimization in reinforcement learning [52]. Stochastic Gradient Descent (SGD) and its variants are driving the practical success of machine learning approaches. Naturally, understanding the limits of performance of SGD in the nonconvex setting has become an important avenue of research in recent years [21, 4, 30, 44, 23, 15, 59].

We are interested in solving the unconstrained *stochastic*, *nonconvex* optimization problem of the form

$$\min_{x \in \mathbb{R}^d} f(x) := \mathbb{E}_{\xi \sim \mathcal{D}} \left[ f_\xi(x) \right], \tag{1}$$

where $f(\cdot)$ is smooth and possibly nonconvex, and $\xi$ is a random vector drawn from a distribution $\mathcal{D}$. Moreover, we are interested in an important special case of (1), when the expectation can be written as the average of $n$ smooth functions:

$$\min_{x \in \mathbb{R}^d} \left[ f(x) := \frac{1}{n} \sum_{i=1}^{n} f_i(x) \right]. \tag{2}$$

For a general nonconvex differentiable objective $f : \mathbb{R}^d \to \mathbb{R}$, finding a global minimum of $f$ is in general intractable [42, 54]. There are two common strategies to analyze optimization methods for nonconvex functions. The first one is to scale down the requirements on the solution of interest from global optimality to some relaxed version, e.g., first-order stationary point. However, such solutions do not exclude the possibility of approaching a suboptimal local minima or a saddle point. Another approach is to study nonconvex problems with additional structural assumption in the hope of

---

[*]First two authors have equal contribution.

[†]École polytechnique fédérale de Lausanne

36th Conference on Neural Information Processing Systems (NeurIPS 2022).

convergence to global solutions. In this direction, several relaxations of convexity have been proposed and analyzed, for instance, star convexity, quasar-convexity, error bounds condition, restricted secant inequality, and quadratic growth [26, 24, 23]. Many of the aforementioned relaxations have limited application in real-world problems.

Recently, there has been a surge of interest in the analysis of functions satisfying the so-called Kurdyka-Łojasiewicz (KŁ) inequality [9, 10]. Of particular interest is the family of functions that satisfy global KŁ inequality. Specifically, we say that $f(\cdot)$ satisfies *(global) KŁ inequality* if there exists some continuous function $\phi(\cdot)$ such that $\|\nabla f(x)\| \geq \phi(f(x) - \inf_x f(x))$ for all $x \in \mathbb{R}^d$. If this inequality is satisfied for $\phi(t) = \sqrt{2\mu}\ t^{1/\alpha}$, then we say that *(global) $\alpha$-PŁ condition* is satisfied for $f(\cdot)$. The special case of KŁ condition, 2-PŁ, often referred as Polyak-Łojasiewicz or PŁ condition, was originally discovered independently in the seminal works of B. Polyak, T. Ležanski and S. Łojasiewicz [48, 33, 38, 39]. Notably, this class of problems has found many interesting emerging applications in machine learning, for instance, policy gradient (PG) methods in reinforcement learning [41, 1, 56], generalized linear models [40], over-parameterized neural networks [2, 57], linear quadratic regulator in optimal control [13, 19], and low-rank matrix recovery [7].

Despite increased popularity of KŁ and $\alpha$-PŁ assumptions, the analysis of stochastic optimization under it remains limited and the majority of works focus on deterministic gradient methods. Indeed until recently, only the special case of $\alpha$-PŁ with $\alpha = 2$ was mainly addressed in the literature [26, 23, 30, 53, 49]. In this paper, we study the sample complexities of stochastic optimization for the broader class of nonconvex functions with global KŁ property.

## 1.1 Related Works and Open Questions

**Stochastic gradient descent.** A plethora of existing works has studied the sample complexity of SGD and its variants for finding an $\epsilon$-stationary point of general nonconvex function $f$, that is, a point $x \in \mathbb{R}^d$ for which $\mathbb{E}[\|\nabla f(\hat{x})\|] \leq \epsilon$. For instance, [21] showed that for a smooth objective (one with Lipschitz gradients) under bounded variance (BV) assumption, SGD with properly chosen stepsizes reaches an $\epsilon$-stationary point with the sample complexity of $\mathcal{O}(\epsilon^{-4})$. Recently, [30, 56] further extended the result under a much milder *expected smoothness* (ES) assumption on stochastic gradient. While this sample complexity is known to be optimal for general nonconvex functions, a naive application of this result to the function value using $\alpha$-PŁ condition would lead to a suboptimal $\mathcal{O}(\epsilon_f^{-4/\alpha})$ sample complexity for finding an $\epsilon_f$-optimal solution, i.e., $\mathbb{E}[f(x) - f^\star] \leq \epsilon_f$. Recently, [20] studied SGD and established convergence rates for $\alpha$-PŁ functions under BV assumption. Their sample complexity result is $\mathcal{O}(\epsilon_f^{-(4-\alpha)/\alpha})$ in our notation. Later [35] considered SGD scheme with random reshuffling under local and global KŁ conditions and provided convergence in the iterates for $\alpha \in (1, 2]$. We note that our proof techniques are different from [20] and [35] and are not limited merely to the case of BV assumption. In this work, we will answer the following open question:

> *What is the exact performance limit of* SGD *under global KŁ condition and a more practical model of stochastic oracle?*

**Variance reduction.** There has been extensive research on development of algorithms which improve the dependence on $n$ and/or $\epsilon$ for both problems (1) and (2) (over simple methods such as SGD and Gradient Descent (GD) ). One important family of techniques[3] is *variance reduction*, which has emerged from the seminal works of Blatt et. al [8]. The main idea of *variance reduction* is to make use of the stochastic gradients computed at previous iterations to construct a better gradient estimator at a relatively small computational cost. Various modifications, generalizations, and improvements of the variance reduction technique appeared in subsequent work, for instance, [50, 28, 16] to name a few.

**Finite-sum case.** A number of recently proposed algorithms such as SNVRG [58], SARAH [45], STORM [15], SPIDER [17], and PAGE [36] achieve the sample complexity $\mathcal{O}\left(n + \frac{\sqrt{n}}{\epsilon^2}\right)$ when minimizing a general nonconvex function with finite sum structure (2). This result is also known to be optimal in this setting [36]. [27] studies SARAH in finite sum case under local KŁ assumption and proves convergence in the iterates. The study in [27] is only asymptotic analysis and the dependence

---

[3]Another independent direction is to make use of higher order information [43, 18, 3].

on the parameters $\kappa$ and $n$, which are important in practice for quantifying the improvement over GD and SGD are ignored. [34] proposes an SVRG-based algorithm for KŁ functions and [55, 45, 46] study other variance reduction techniques, but they only analyze the special case $\alpha = 2$. Under 2-PŁ condition, these methods further improve to $\mathcal{O}\big( (n + \kappa\sqrt{n}) \log(\frac{1}{\epsilon_f}) \big)$ sample complexity[4] for finding an $\epsilon_f$-optimal solution. However, it is not clear if it is possible to provide any non-asymptotic guaranties for variance reduced methods under $\alpha$-PŁ condition for any $\alpha \in [1, 2)$. In our work, we will answer the following open question:

> *What is the extent of improvement any variance reduction scheme can provide under global $\alpha$-PŁ condition for finite-sum objectives of the form* (2)*?*

**Online/streaming case.** While variance reduction methods have been initially designed for problems of the form (2), it was later discovered that they also improve over SGD when solving (1) [32, 37][5]. The analysis of these methods was obtained for *general nonconvex* functions (for minimizing the norm of the gradient, $\mathbb{E}\left[\|\nabla f(\hat{x})\|\right] \leq \epsilon$) and later extended to 2-*PŁ objectives* for minimizing the function value, $\mathbb{E}\left[f(x) - f^\star\right] \leq \epsilon_f$. For example, the methods in [58, 45, 17, 36] achieve $\mathcal{O}(\epsilon^{-3})$ complexity improving over $\mathcal{O}(\epsilon^{-4})$ complexity of SGD for finding an $\epsilon$-stationary point. Under the 2-PŁ condition, these results can be extended to global convergence with $\mathcal{O}(\epsilon_f^{-1})$ sample complexity [36]. However, in contrast to a general nonconvex case, variance reduction under 2-PŁ assumption does not show any improvement over SGD in terms of $\epsilon_f$. We highlight that all existing analysis of variance reduction under $\alpha$-PŁ condition *is restricted only to a special case $\alpha = 2$*. We refer the reader to Appendix C, where we elaborate on the key difficulties in the analysis for the cases $\alpha \in [1, 2)$. Since the direct analysis for $\alpha \in [1, 2)$ is challenging, in order to obtain the global convergence in this setting, one could naively translate the complexity for finding a stationary point of a general nonconvex function (which is $\mathcal{O}(\epsilon^{-3})$) to convergence in a function value by using $\alpha$-PŁ condition: $\sqrt{2\mu} \left(f(\hat{x}) - f^\star\right)^{1/\alpha} \leq \|\nabla f(\hat{x})\|$. This would result in $\mathcal{O}(\epsilon_f^{-3/\alpha})$ sample complexity. However, there are two serious issues with this approach. First, this complexity does not provide any improvement over SGD in the most interesting practical case $\alpha = 1$ and gives strictly worse result for all $\alpha > 1$. Second, the guarantees for general nonconvex optimization hold on average, in the sense that the point $\hat{x}$ is sampled uniformly from all the iterates of the algorithm. It would be more desirable to instead derive last iterate convergence guarantees under KŁ ($\alpha$-PŁ) condition. In this work, we will address the following open question:

> *Is it possible to accelerate the $\mathcal{O}\big(\epsilon_f^{-(4-\alpha)/\alpha}\big)$ sample complexity of* SGD *under global $\alpha$-PŁ condition for stochastic objectives of the form* (1)*?*

## 1.2 Contributions

In this work, we provide an extensive analysis of stochastic optimization under global KŁ condition and answer all the above questions. More precisely, our contributions are as follows

- We provide a new framework for the analysis of the dynamics of SGD under global KŁ condition (see Section 3). It is based on the analysis of SGD dynamic which is governed by a recursive inequality (see Equation (6)). As a result of this analysis, we introduce a set of conditions (see Theorem 1) for designing proper stepsizes to guarantee convergence.

- Using this framework, we provide sharp analysis of SGD under a general ES assumption (Assumption 4) and demonstrate that the sample complexity $\mathcal{O}(\epsilon_f^{-(4-\alpha)/\alpha})$ is tight for the dynamical system describing SGD.

- Next, we propose PAGER, a new variance reduction scheme with parameter restart. A carefully chosen sequence of parameters of PAGER allows the algorithm to adapt to the nonconvex geometry of the problem and establish state-of-the-art convergence guarantees for minimizing $\alpha$-PŁ functions. In online setting (1), we obtain $\mathcal{O}\left(\epsilon_f^{-2/\alpha}\right)$ sample complexity of PAGER, which beats $\mathcal{O}\left(\epsilon_f^{-(4-\alpha)/\alpha}\right)$ complexity of SGD for the whole spectrum of

---

[4]$\kappa = \mathcal{L}/\mu$ is the analogue of condition number, $\mathcal{L}$ is defined in Assumption 6.

[5]Under additional assumptions such as smoothness of individual functions $f_\xi(\cdot)$ or even milder condition such as *average L-smoothness* (Assumption 6).

parameters $\alpha \in [1, 2)$. In particular, for the important special case of 1-PŁ, this leads to the first optimal algorithm with $O(\epsilon_f^{-2})$ sample complexity, which already matches with the lower bound known for stochastic convex optimization [42].

- Furthermore, we obtain faster rates with PAGER in finite sum case (2), providing the first acceleration over GD and SGD under $\alpha$-PŁ condition.

In Table 1, we summarize the sample complexity results for stochastic optimization under $\alpha$-PŁ and BV assumptions. We also establish sharp convergence results for convergence in the iterates to the set $X^\star$ of optimal points and provide a summary in Table 2 in the Appendix.

Table 1: Summary of sample complexity results for $\alpha$-PŁ functions (Assumption 3) under average $\mathcal{L}$-smoothness (Assumptions 6) and bounded variance (Assumptions 5). Quantities: $\alpha$ = PŁ power; $\mu$ = PŁ constant; $\kappa = \mathcal{L}/\mu$; $\sigma^2$ = variance. The entries of the table show the expected number of stochastic gradient calls to achieve $\mathbb{E}\left[f(x_k) - f^\star\right] \leq \epsilon_f$ .

| Method | Finite sum case | Online case |
|---|---|---|
| GD | $\mathcal{O}\left(n\kappa \left(\frac{1}{\epsilon_f}\right)^{\frac{2-\alpha}{\alpha}}\right)$ | N/A |
| SGD | $\mathcal{O}\left(\frac{\kappa\sigma^2}{\mu} \left(\frac{1}{\epsilon_f}\right)^{\frac{4-\alpha}{\alpha}}\right)$ | $\mathcal{O}\left(\frac{\kappa\sigma^2}{\mu} \left(\frac{1}{\epsilon_f}\right)^{\frac{4-\alpha}{\alpha}}\right)$ |
| PAGER | $\widetilde{\mathcal{O}}\left(n + \sqrt{n}\kappa \left(\frac{1}{\epsilon_f}\right)^{\frac{2-\alpha}{\alpha}}\right)$ (new) | $\mathcal{O}\left(\left(\frac{\sigma^2}{\mu} + \kappa^2\right) \left(\frac{1}{\epsilon_f}\right)^{\frac{2}{\alpha}}\right)$ (new) |

## 2 Assumptions and Discussion

In this section, we introduce the assumptions we make throughout the paper.

**Assumption 1.** *The gradient of $f(\cdot)$ is Lipschitz continuous, that is, for all $x$ and $y$, $\|\nabla f(x) - \nabla f(y)\| \leq L \|x - y\|$, $L > 0$ is referred to as the Lipschitz constant.*

Furthermore, we assume that the objective function $f$ is lower bounded, i.e., $f^* := \inf_x f(x) > -\infty$, and it satisfies the following inequality

**Assumption 2** (global KŁ or global Kurdyka-Łojasiewicz). *Let $\phi : \mathbb{R}^+ \to \mathbb{R}^+$ be a continuous function such that $\phi(0) = 0$ and $\phi^2(\cdot)$ is convex. The function $f(\cdot)$ is said to satisfy global Kurdyka-Łojasiewicz inequality if*

$$\|\nabla f(x)\| \geq \phi\left(f(x) - f^*\right) \quad \text{for all } x \in \mathbb{R}^d. \tag{3}$$

**Assumption 3** ($\alpha$-PŁ or Polyak-Łojasiewicz). *There exists $\alpha \in [1, 2]$ and $\mu > 0$ such that*

$$\|\nabla f(x)\|^\alpha \geq (2\mu)^{\alpha/2} \left(f(x) - f^*\right) \quad \text{for all } x \in \mathbb{R}^d. \tag{4}$$

*We refer to $\alpha$ as the PŁ power and $\mu$ as the PŁ constant.*

It is straightforward to see that the $\alpha$-PŁ is a special case of KŁ with $\phi(t) = \sqrt{2\mu}\, t^{1/\alpha}$.

**Connections with other assumptions.** Another commonly adopted way to define the global KŁ property is to assume that $\rho'\left(f(x) - f^\star\right) \cdot \|\nabla f(x)\| \geq 1$ for all $x \in \mathbb{R}^d$, where $\rho(t)$ is called a disingularizing function and $\rho'(\cdot)$ denotes its derivative. Moreover, disingularizing function satisfies the following conditions, it is continuous, concave, $\rho(0) = 0$, and $\rho'(t) > 0$ [35].

If the above assumption holds for $\rho(t) := \frac{1}{\theta}t^\theta$ and $\theta > 0$, then Assumption 3 is satisfied with PŁ power $\alpha = \frac{1}{1-\theta}$. However, $\alpha$-PŁ condition is more general since it allows to consider the case $\alpha = 1$. For example, consider the function of one variable $f(x) = (e^x + e^{-x})/2 - 1$, then $|f'(x)| \geq f(x)$ for all $x$. Thus $f(x)$ satisfies Assumption 3 with $\alpha = 1$ and $\mu = 1/2$. Moreover, this function is convex, but it does not satisfy inequality $\rho'\left(f(x) - f^\star\right) \cdot \|\nabla f(x)\| \geq 1$ for any choice of $\rho(t)$.

We also provide several non-convex problems for which $\alpha$-PŁ holds with $\alpha \in [1, 2]$ in the Appendix A. Other forms of $\phi(t)$ also appear in practice, e.g., squared cross entropy loss function satisfies the KL condition with $\phi(t) = \min\{t, \sqrt{t}\}$, [51].

The intuition behind the special case $\alpha = 1$ is that the function is allowed to be flat near the set of optimal points $X^\star = \arg\min_x f(x)$.

**Assumption 4** ($k$-ES, Expected Smoothness of order $k$)**.** *The stochastic gradient estimator $g_k(x, \xi)$ is an unbiased estimate of the gradient $\nabla f(x)$ at any given point $x$ and its second moment satisfies*

$$\mathbb{E}\left[\|g_k(x, \xi)\|^2\right] \leq 2A \cdot h\big(f(x) - f^*\big) + B \cdot \|\nabla f(x)\|^2 + \frac{C}{b_k}, \quad \text{for all } x \in \mathbb{R}^d, \tag{5}$$

*where $A, B, C$ are non-negative constants. $h : \mathbb{R}^+ \to \mathbb{R}^+$ is a concave continuously differentiable function with $h'(t) \geq 0$, $h(0) = 0$. The expectation is taken over random vector $\xi \sim \mathcal{D}$. We call $b_k$ the cost of such estimator.*

This assumption encompasses previous assumptions in the literature. For instance, it is straightforward to see that an estimator satisfies the standard *bounded variance assumption* [21] when $h(t) = 0$, $B = 1$, and $b_k = 1$ in (5). Gradient estimators with *relaxed growth assumption* [6, 11] are also special cases of (5) for $h(t) = 0$ and $b_k = 1$. A closely related assumption to the relaxed growth was introduced in [53] which holds when $h(t) = 0$ and $C = 0$ in (5). *Expected smoothness assumption* [30, 24, 56] is the closest assumption to $k$-ES and it holds when $h(t) = t$ and $b_k = 1$. Notably, ES assumption is satisfied in practical scenarios such as mini-batching, importance sampling and compressed communication [30]. More recently, it has been shown that PG method with softmax policies and log barrier regularization can be modeled using ES assumption [56]. Note that due to the first term in (5), i.e., $2A \cdot h(f(x) - f^*)$, the second moment can be large when the objective gap at $x$ is large. Such property is not captured by standard *bounded variance* (Assumption 5). The flexibility and advantages of introducing such additional term are elaborated in detail in the literature [24, 30, 25, 56].

We highlight two special cases for the sequence $b_k$: $b_k = \Theta(k^\tau)$ with $\tau \geq 0$ and $b_k = \Theta(q^k)$ with $q > 1$. For example, Monte Carlo sampling and mini-batching allow us to design estimators with such $\{b_k\}_{k \geq 0}$ sequences. When a gradient estimator satisfies Assumption 4, unless $b_k$ is bounded for all $k$, it essentially means that we have a mechanism to reduce its variance. More precisely, the variance decreases according to the sequence $1/b_k$. As we show in Section 4, if such estimator exists, it results in a better convergence rate compared to vanilla SGD and the improvement is captured by sequence $b_k$. On the other hand, usually the access to such estimator comes with a *cost* proportional to $b_k$, e.g., mini-batch setting described in Section 4. Thus we refer to $b_k$ as the *cost* of the estimator $g_k(x, \xi)$.

## 3 Stochastic Gradient Method

Algorithm 1 summarizes the steps of a slightly modified SGD which we analyze in this work. We call this algorithm SGD with restarts.[6] This algorithm updates the point $x$ for $T$ number of iterations within an inner-loop. Note that, in the inner-loop, the step-size remains unchanged and the iterates are updated via $x_{t+1} = x_t - \eta g_k(x_t, \xi_t)$, where $\eta$ is the step-size and $\{\xi_t\}_{t \geq 0}$ are independent random vectors. The cost $b_k$ of the gradient estimator $g_k(x, \xi)$ remain the same within the inner loop of Algorithm 1.

---
**Algorithm 1:** SGD with restarts
---
1: Initialization: $x, T, K, \{\eta_k : k = 0, ..., K - 1\}$
2: **for** $k = 0, \ldots, K - 1$ **do**
3:    $\eta \leftarrow \eta_k$
4:    **for** $t = 0, \ldots, T - 1$ **do**
5:       $x \leftarrow x - \eta g_k(x, \xi_t)$
6: **return** $x$

---

### 3.1 Dynamics of SGD

Let $\{x_t\}_{t \geq 0}$ be the sequence of points generated by the inner loop of Algorithm 1, and Assumptions 1, 2, 4 are satisfied. Then the dynamics of SGD in the inner-loop of Algorithm 1 is characterized by

---

[6]Note that if we set $K = 1$, then Algorithm 1 reduces to SGD with constant step-size.

**Lemma 1.** *Under Assumptions 1, 2, and 4 with constant cost, i.e., $b := b_k$, we obtain*

$$\delta_{t+1} \leq \delta_t + a\eta^2 \cdot h(\delta_t) - \frac{\eta}{2}\phi^2(\delta_t) + \frac{d\eta^2}{b}, \tag{6}$$

*where $\delta_t := \mathbb{E}\left[f(x_t) - f^\star\right]$, $a := LA$, $d := \frac{LC}{2}$, $\eta := \eta_k$.*

Understanding the dynamics of this recursion, allows us to establish the global convergence of SGD. Our approach consists of two main steps: i) Finding the stationary[7] point of (6) when the inequality is replaced by equality and for a fixed step-size, $\eta_k = \eta$, which we denote by $r(\eta)$. ii) Selecting the step-sizes $\{\eta_k\}$ and sequence $\{b_k\}_{k\geq0}$, such that the corresponding stationary points $\{r(\eta_k)\}_{k\geq0}$ (defined below) converge to zero as $k$ increases.

The stationary point of (6) after replacing inequality with equality must satisfy the equation:

$$a\eta^2 h(t) + \frac{d\eta^2}{b} = \frac{\eta}{2}\phi^2(t). \tag{7}$$

Let us call this stationary point $r(\eta)$. To complete the first step, we approximate $r(\eta)$ by a polynomial function of $\eta$. In other words, we find $\nu \in \mathbb{R}^+$ such that $r(\eta) = \Theta(\eta^\nu)$.

For the second step of our framework, we should design the stepsizes. Next result introduces a set of conditions that allow us to design the stepsizes, which will guarantee convergence. The detailed derivations are presented in the Appendix.

**Theorem 1.** *Suppose there exist $\nu \geq 0$, $\{\omega_j\}_{j\geq0}$, and $\zeta \geq 0$ such that $\eta_k = \Theta(k^{-\zeta})$, $r(\eta_k) = \Theta(k^{-\zeta\nu})$, $|1 - \omega_k| < 1$, and*

$$1 + a\eta_k^2 h'\big(r(\eta_k)\big) - \eta_k\phi'\big(r(\eta_k)\big)\phi\big(r(\eta_k)\big) = 1 - \omega_k k^{-1}. \tag{8}$$

*Then, $\delta_k = \mathcal{O}(k^{-\zeta\nu})$ and the iteration complexity of Algorithm 1 with $T = \Omega(1/\min_j \omega_j)$ is $\mathcal{O}(\epsilon_f^{-1/(\zeta\nu)})$.*

As a consequence of Theorem 1, we present the iteration complexity of SGD for $\alpha$-PŁ functions.

**Corollary 1.** *Consider a special case of Assumption 4 with $h(t) = t^\beta$ and $b_k = k^\tau$, where $\beta \in (0,1]$ and $\tau \geq 0$. Suppose the objective function $f$ satisfies Assumptions 1 and 3. Let $\gamma := \alpha\beta$. Then, for any $\epsilon_f > 0$, Algorithm 1 returns a point $x$ with $\mathbb{E}\left[f(x) - f^\star\right] \leq \epsilon_f$ after $N := K \cdot T$ iterations.*

*i) If $\gamma = 2$ ($\alpha = 2$ and $\beta = 1$), we have*

$$N = \mathcal{O}(\epsilon_f^{-\frac{1}{1+\tau}}), \text{ with } \eta_k = \Theta(k^{-1}).$$

*ii) If $\gamma < 2$, we have*

$$N = \mathcal{O}\big(\epsilon_f^{-\frac{4-\alpha}{\alpha(\tau+1)}}\big) \text{ with } \eta_k = \Theta\big(k^{-\frac{\tau+1}{2-\alpha/2}+\tau}\big) \text{ if } \tau \leq \frac{\gamma}{4-\alpha-\gamma}, \text{ and}$$

$$N = \mathcal{O}\big(\epsilon_f^{-\frac{4-\alpha-\gamma}{\alpha}}\big) \text{ with } \eta_k = \Theta\big(k^{-\frac{2-\gamma}{4-\alpha-\gamma}}\big) \text{ if } \tau > \frac{\gamma}{4-\alpha-\gamma}.$$

To verify the above result empirically, we simulated $\delta_t$ in (6) throughout all iterations of Algorithms 1 for different sets of parameters and presented the results in Figure 1 along with their corresponding convergence rates given in Corollary 1. As it is shown in these figure, the above convergence rates correctly capture the behaviour of the dynamics in (6). As an example, in Figure 1(b), the red solid curve shows the rate of $\delta_k$, i.e., $\log(\delta_k)$ as a function of $\log(k)$ for $\gamma = 1.1, \alpha = 1.4$, and $\tau = 0.9$. Based on Corollary 1, the convergence rate of Algorithm 1 for this setting is $\mathcal{O}(\epsilon_f^{-0.98})$ or equivalently $\log(\delta_k) = (-\frac{\alpha(\tau+1)}{4-\alpha})\log(k) \approx -1.02\log(k)$ which is shown by red dashed line. Next, we discuss how the results in Corollary 1 generalizes the existing work in the literature.

**Comparison to related works.** Authors in [30] studied the convergence of SGD for 2-PŁ objectives, under a stronger assumption than Assumption 4. More precisely, they assumed an estimator that satisfies Assumption 4 with $\tau = 0$ and $h(t) = t$ and obtained the convergence rate of $\mathcal{O}(\epsilon_f^{-1})$. This

---

[7]Stationary point of a dynamic is its convergence point.

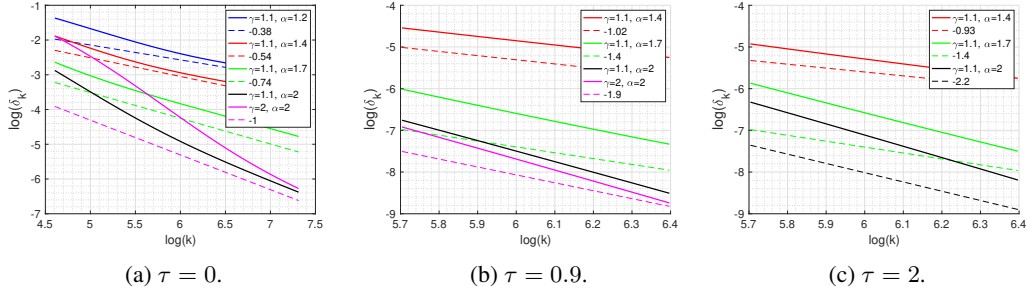

Figure 1: Behavior of the dynamics in (6) for $h(t) = t^\beta$, $\phi(t) = \sqrt{2\mu}\, t^{1/\alpha}$, $\tau \in \{0, 0.9, 2\}$, and different $\alpha, \beta$. Each solid line shows $\log(\delta_k)$ as a function of $\log(k)$, for a given set of parameters and each dashed line shows the corresponding theoretical convergence rate of $\delta_k$ presented in Corollary 1. The numbers assigned to dashed lines indicate the slope of those lines. (a) and (b) verify the case corresponding to $\tau \le \gamma/(4 - \alpha - \gamma)$ and (c) verifies the case $\tau > \gamma/(4 - \alpha - \gamma)$. Note that the distance between the dashed and solid lines is due to constant factors.

is consistent with our rate presented in the Corollary 1. It is worth noting that in this setting, $\mathcal{O}(\epsilon_f^{-1})$ is optimal [30].

The authors in [56] studied the performance of SGD for 1-PŁ objectives. Assuming that the gradient estimator satisfies Assumption 4 with $\tau = 0$ and $h(t) = t$, they obtain $\mathcal{O}(\epsilon_f^{-3})$ sample complexity. This result can be recovered from Corollary 1 by setting $\tau = 0, \gamma = 1$, and $\alpha = 1$. Note that in this case, the cost of each iteration is $b_k = 1$, which means that the iteration complexity coincides with the sample complexity. We note that our proof technique is different than in [56] and allows to consider more general assumptions. Finally, [20] studied SGD with $\alpha$-PŁ objectives for $\alpha \in [1, 2]$ under bounded variance Assumption 5 with $b_k = 1$ and obtained similar convergence rate to ours. We recover their result as a special case by setting $A = 0$ and $\tau = 0$ in Corollary 1, if we set $T = 1$ we also recover the same (up to a constant) step-sizes $\eta_k = \Theta\left(k^{-\frac{2}{4-\alpha}}\right)$. However, we highlight that our proof technique is different from [20], and more generic since it holds for a general Assumption 4.

### 3.2 Sample complexity of SGD

The result of Corollary 1 suggests that by increasing the cost of the gradient estimator $b_k$ over the iterations, one can achieve a better iteration complexity of Algortihm 1. In particular, it improves with $\tau$ until it reaches the minimum $N = \mathcal{O}\left(\epsilon_f^{-\frac{4-\alpha-\gamma}{\alpha}}\right)$ at $\tau = \frac{\gamma}{4-\alpha-\gamma}$ and does not change for larger values of $\tau$. However, we are merely interested in the iteration complexity in practice, since the computational cost at each iteration can be prohibitively large. A more adequate measure is the total computational cost (sample complexity) of the method. It is interesting whether increasing $b_k$ over the iterations may also result in a better sample complexity for finding an $\epsilon_f$-optimal solution, than for the constant choice, e.g., $b_k = 1$. The following lemma shows the contrary.

**Proposition 1.** *Let the assumptions of Corollary 1 hold, $b_k = \Theta\left(k^\tau\right)$, $T = \Theta\left(1\right)$. Then the expected total computational cost (sample complexity) of Algorithm 1 is*

$$
cost := T \cdot \sum_{k=0}^{K-1} b_k = \begin{cases} \mathcal{O}\left(\epsilon_f^{-\frac{4-\alpha}{\alpha}}\right) & \text{for } 0 \le \tau \le \frac{\gamma}{4-\alpha-\gamma}, \\ \mathcal{O}\left(\epsilon_f^{-\frac{(4-\alpha-\gamma)(\tau+1)}{\alpha}}\right) & \text{for } \tau > \frac{\gamma}{4-\alpha-\gamma}. \end{cases}
$$

The above result implies that increasing the cost of the gradient estimator with iterations does not improve the total sample complexity of Algorithm 1. Therefore, one can simply select $b_k = 1$ ($\tau = 0$) and obtain $\mathcal{O}\left(\epsilon_f^{-\frac{4-\alpha}{\alpha}}\right)$ sample complexity.

### 3.3 Tightness of rates in Corollary 1

In this section, we show that when $\tau = 0$, the convergence rates presented in Corollary 1 are tight for the dynamic (6) describing the progress of SGD. More precisely, if there exists a function $f$ and a

gradient estimator satisfying the assumptions in Corollary 1 such that its corresponding recursive inequality (6) is an equality, then its convergence rate, presented in Corollary 1 cannot be improved by any choices of stepsizes $\{\eta_k\}_{k \geq 0}$. Next proposition summarizes our results about the tightness of our convergence rates in Corollary 1.

**Proposition 2.** *Consider the following recursion*

$$\delta_{k+1} = \delta_k + a\eta_k^2 \cdot h(\delta_k) - \frac{\eta_k}{2}\phi^2(\delta_k) + \frac{d\eta_k^2}{b_k}, \quad \text{for all } k \geq 0,$$

*where $a \geq 0$, $d > 0$, $h(t) = t^\beta$ with $\beta \in (0,1]$, $\phi(t) = \sqrt{2\mu}t^{1/\alpha}$ with $\alpha \in [1,2]$, and $b_k = \Theta(1)$. Then $\delta_k = \Omega(k^{-\frac{\alpha}{4-\alpha}})$ for any sequence of $\{\eta_k\}_{k \geq 0}$. Moreover, this rate is achieved by the choice $\eta_k = \Theta(k^{-\frac{1}{2-\alpha/2}})$.*

# 4   Faster Rates with Variance Reduction

---

**Algorithm 2:** PAGER (PAGE with restarts)

---

1: Initialization: $\bar{x}_0, \bar{g}_0, K, \{\Lambda_k = (\eta_k, T_k, p_k, b_k, b'_k) : k = 0, ..., K-1\}$
2: **for** $k = 0, \dots, K-1$ **do**
3:     $(x_0, g_0) \leftarrow (\bar{x}_k, \bar{g}_k)$
4:     $(\eta, p, b, b') \leftarrow (\eta_k, p_k, b_k, b'_k)$
5:     **for** $t = 0, \dots, T_k - 1$ **do**
6:         $x_{t+1} = x_t - \eta g_t$
7:         Sample $\chi \sim \text{Bernoulli}(p)$
8:         **if** $\chi = 1$ **then**
9:             $g_{t+1} = \frac{1}{b}\sum_{i=1}^{b} \nabla f_{\xi_{t+1}^i}(x_{t+1})$
10:        **else**
11:            $g_{t+1} = g_t + \frac{1}{b'}\sum_{i=1}^{b'} \nabla f_{\xi_{t+1}^i}(x_{t+1}) - \frac{1}{b'}\sum_{i=1}^{b'} \nabla f_{\xi_{t+1}^i}(x_t)$
12:     $(\bar{x}_{k+1}, \bar{g}_{k+1}) \leftarrow (x_{t+1}, g_{t+1})$
13: **Return:** $\bar{x}_K$

---

To simplify the exposition of the results in this section, let us assume that $g_k(x_t, \xi_t)$ is constructed explicitly via mini-batching $g_k(x_t, \xi_t) := \frac{1}{b_k}\sum_{i=1}^{b_k} \nabla f_{\xi_t^i}(x_t)$, where $\xi_t := \left(\xi_t^1, \dots, \xi_t^{b_k}\right)$ is a random vector of independent entries, $\xi_t$ are independent for all iterations, $\{\nabla f_{\xi_t^i}(x_t)\}_{i=1}^{b_k}$ are queries provided by an oracle such that $\mathbb{E}[\nabla f_{\xi_t^i}(x_t)] = \nabla f(x_t)$ and $\mathbb{E}[\|\nabla f_{\xi_t^i}(x_t) - \nabla f(x_t)\|^2] \leq \sigma^2$ for all $t \geq 0$. The variance of this estimator diminishes linearly in the size of the mini-batch $b_k$, i.e., $g_k(x_t, \xi_t)$ satisfies

**Assumption 5** ($k$-BV, bounded variance)**.** *Let Assumption 4 hold with $A = 0$, $B = 1$ and $C = \sigma^2$, i.e., $\mathbb{E}\left[\|g_k(x, \xi) - \nabla f(x)\|^2\right] \leq \frac{\sigma^2}{b_k}$.*

Additionally, we assume that we have access to a gradient estimator $g'_k(x, \xi)$, which satisfies the following

**Assumption 6** (Average $\mathcal{L}$-smoothness (of order $k$))**.** *Let $g'_k(x, \xi) := \frac{1}{b'_k}\sum_{i=1}^{b'_k} \nabla f_{\xi^i}(x)$ and $g'_k(y, \xi) := \frac{1}{b'_k}\sum_{i=1}^{b'_k} \nabla f_{\xi^i}(y)$ be unbiased mini-batch estimators of the gradient of $f(\cdot)$ at points $x$ and $y$, respectively for shared stochasticity $\xi^i \sim \mathcal{D}$ for each $i = 1, \dots, b'_k$ and $\xi = (\xi^1, \dots, \xi^{b_k})$. Define $\widetilde{\Delta}(x, y) := g'_k(x, \xi) - g'_k(y, \xi)$. The average $\mathcal{L}$-smoothness (of order $k$) holds if there exists $\mathcal{L} \geq 0$ such that $\mathbb{E}\left[\left\|\widetilde{\Delta}(x, y) - \Delta(x, y)\right\|^2\right] \leq \frac{\mathcal{L}^2}{b'_k}\|x - y\|^2$ for all $x, y \in \mathbb{R}^d$, where $\Delta(x, y) := \nabla f(x) - \nabla f(y)$.*

**Remark 1.** *The Assumption 6 holds in several standard settings. For instance, if each $\nabla f_{\xi^i}(x)$ is Lipschitz with constant $\bar{L}$ (almost surely or on average), then Assumption 6 holds with $\mathcal{L} \leq \bar{L}$. Another example is when $f(\cdot)$ is of the form (2) and $b'_k = n$, then $\mathcal{L} = 0$.*

## 4.1 PAGER – a new variance reduction for $\alpha$-PŁ objectives

We remark from the analysis of Algorithm 1 in Section 3 that merely playing with choice of $\eta_k$ and $b_k$ (chosen as polynomial functions of $k$) is not sufficient to improve the convergence, hence, we need to construct more sophisticated gradient estimator and reduce the variance using control variate. Now, we highlight the main algorithmic ingredients of our construction. First, let us describe the variance reduced estimator named PAGE, which will be the main building block for our Algorithm 2. PAGE was introduced and analyzed in [36] and is known to be optimal for finding a first order stationary point. Moreover, it is easy to implement and designed via a small modification to mini-batch SGD

$$g_{t+1} = \begin{cases} \frac{1}{b}\sum_{i=1}^{b} \nabla f_{\xi_{t+1}^i}(x_{t+1}), & \text{w.p.} \quad p, \\ g_t + \frac{1}{b'}\sum_{i=1}^{b'}\left(\nabla f_{\xi_{t+1}^i}(x_{t+1}) - \nabla f_{\xi_{t+1}^i}(x_t)\right), & \text{w.p. } 1-p, \end{cases}$$

where $p$ is a small probability and mini-batch sizes satisfy $b > b'$.

However, while the method looks simple, the extension of its analysis to $\alpha$-PŁ functions faces several difficulties. [8] Therefore, we introduce a new method, which we call PAGER (Algorithm 2) – a *Probabilistic Average Gradient Estimator with parameter Restart*. It takes as input the sequence of parameters $\{\Lambda_k := (\eta_k, T_k, p_k, b_k, b'_k) : k = 0, ..., K-1\}$, where $T_k$ is the length of stage $k$, $\eta_k, p_k, b_k, b'_k$ step-size, probability, and batch-sizes at stage $k$. PAGER updates this sequence of parameters in the outer loop $k = 0, \ldots, K-1$ and applies PAGE estimator with a fixed set of parameters in the inner loop $t = 0, \ldots, T_k - 1$. We will select $\{\Lambda_k\}_{k\geq 0}$ depending on the PŁ power $\alpha$ to capture the dependence on the geometry of the problem and establish fast rates for each $\alpha$ in settings (1) and (2).

## 4.2 Online case

We present convergence guarantees for Algorithm 2 in the setting (1) and defer its formal proof to Appendix C.

**Theorem 2.** *Let $f(\cdot)$ have the form* (1) *and satisfy Assumptions 1, 3 (with $\alpha \in [1, 2)$), 5 and 6, let the sequences[9] in Algorithm 2 be chosen as $b'_k = \Theta\big(2^{\frac{(2-\alpha)k}{\alpha}}\big)$, $p_k = \Theta\big(2^{\frac{-(2-\alpha)k}{\alpha}}\big)$, $b_k = \Theta\big(2^{\frac{2k}{\alpha}}\big)$, $T_k = \Theta\big(2^{\frac{(2-\alpha)k}{\alpha}}\big)$, $\eta_k = \Theta\big(1\big)$. Then, for any $\epsilon_f > 0$ Algorithm 2 returns a point $x$ with $\mathbb{E}\left[f(x) - f^\star\right] \leq \epsilon_f$ after $N := \sum_{k=0}^{K-1} T_k = \mathcal{O}\big(\kappa \epsilon_f^{-\frac{2-\alpha}{\alpha}}\big)$ iterations, where $\kappa = \mathcal{L}/\mu$. The expected total computational cost (sample complexity) is $\mathcal{O}\left(\left(\frac{\sigma^2}{\mu} + \kappa^2\right)\epsilon_f^{-\frac{2}{\alpha}}\right)$.*

**Improvement over SGD.** Theorem 2 implies that PAGER improves the sample complexity of SGD from $\mathcal{O}\big(\epsilon_f^{-\frac{4-\alpha}{\alpha}}\big)$ to $\mathcal{O}\big(\epsilon_f^{-\frac{2}{\alpha}}\big)$ under $\alpha$-PŁ condition for the whole spectrum of parameters $\alpha \in [1, 2)$. In the case $\alpha = 1$, which holds in many interesting applications (see Appendix A for examples), this leads to $\mathcal{O}\big(\epsilon_f^{-2}\big)$ sample complexity compared to the best known $\mathcal{O}\big(\epsilon_f^{-3}\big)$ for SGD.

**Relation to convex optimization and last iterate convergence.** As a consequence of our analysis we obtain *the optimal sample complexity for convex stochastic optimization* under the additional assumption that the iterates of the method remain bounded, i.e., $\|x_t - x^\star\|^2 \leq D$ for all $t \geq 0$, where $x^\star \in \arg\min_x f(x)$.[10] For 1-PŁ objectives, PAGER has $\mathcal{O}\left(\epsilon_f^{-2}\right)$ sample complexity. Since the iterates of the algorithm are bounded, convexity $\langle\nabla f(x), x - x^\star\rangle \geq f(x) - f(x^\star)$ implies 1-PŁ with $\mu = \frac{1}{2D}$. This observation implies convergence of PAGER for convex objectives with $\mathcal{O}\left(\epsilon_f^{-2}\right)$ sample complexity, which is known to be non-improvable for convex stochastic optimization [42].

---

[8]We refer the reader to Appendix C, where we explain the challenges in the analysis of variance reduction under $\alpha$-PŁ condition and show how we overcome these difficulties using the restart strategy.

[9]For brevity, in Theorem 2 we define the input sequences up to constants hidden in $\Theta(\cdot)$ notation. In fact, our analysis allows to specify these constants and we present detailed derivations in Appendix C.

[10]Note that this assumption is mild since it holds for the iterates of PAGER, for example, if we additionally assume that $f(\cdot)$ is coercive, i.e., $f(x) \to \infty$ for $x \to \infty$.

Moreover, we highlight that this result holds for the *last iterate* of PAGER, while the standard analysis of first order methods for convex functions guarantees convergence for the average iterate [31]. The last iterate convergence for convex objectives was only recently established for SGD by following an involved analysis with a careful control of iterates via suffix-averaging scheme [20].

### 4.3 Finite sum case

Let $f(\cdot)$ have the finite sum form (2). Then we obtain the following result.

**Theorem 3.** *Let $f(\cdot)$ have the form (2) and satisfy Assumptions 1, 3 (with $\alpha \in [1, 2)$) and 6, let the sequences be chosen as $p_k = \frac{1}{n+1}$, $b'_k = 1$, $b_k = n$, $T_k = \Theta\left(2^{\frac{(2-\alpha)k}{\alpha}}\right)$, $\eta_k = \Theta(1)$. Then, for any $\epsilon_f > 0$, Algorithm 2 returns a point $x$ with $\mathbb{E}\left[f(x) - f^\star\right] \le \epsilon_f$ after $N := \sum_{k=0}^{K-1} T_k = \widetilde{\mathcal{O}}\left(n + \sqrt{n}\kappa\epsilon_f^{-\frac{2-\alpha}{\alpha}}\right)$ iterations, where $\kappa = {}^{\mathcal{L}}/_\mu$ The expected total computational cost (sample complexity) is $\widetilde{\mathcal{O}}\left(n + \sqrt{n}\kappa\epsilon_f^{-\frac{2-\alpha}{\alpha}}\right)$.*

The proof is deferred to Appendix C. Theorem 3 quantifies the improvement of PAGER over GD in the finite sum setting in terms of $n$ and over SGD in terms of $\epsilon_f$, see Table 1 for comparison. Recall that GD has sample complexity $\mathcal{O}\left(n\kappa\epsilon_f^{\frac{-(2-\alpha)}{\alpha}}\right)$. When $n$ is large, we get the improvement of order $\sqrt{n}$. Notice that in the limit $\alpha \to 2$, it matches the best known result for 2-PŁ objectives [36].

## 5 Conclusion

We analyzed the complexity of SGD when the objective satisfies global KŁ inequality and the queries from stochastic gradient oracle satisfy weak expected smoothness. We introduced a general framework for this analysis which resulted in a sample complexity of $\mathcal{O}(\epsilon^{-(4-\alpha)/\alpha})$ for SGD with objectives satisfying $\alpha$-PŁ condition. We also demonstrated the tightness of this rate under the specific choice of stepsizes. Last but not least, we developed a modified SGD with variance reduction and restarting (PAGER), which improves the sample complexity of SGD for the whole spectrum of parameters $\alpha \in [1, 2)$ and achieves the optimal rate for the important case of 1-PŁ objectives.

## Acknowledgements

We would like to thank Anas Barakat and Anastasia Kireeva for valuable discussions. This work was supported by ETH AI Center doctoral fellowship, ETH Research Grant funded through the ETH Zurich Foundation, and NCCR Automation funded through the Swiss National Science Foundation.

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
