# Contents

# Appendix

## A  Examples

### A.1  $\alpha$-PŁ Functions

In this section, we provide some examples and applications of global KŁ functions. Particularly, we focus on the class of $\alpha$-PŁ functions with $\alpha \in [1, 2]$. We start with simple one dimensional functions.

**Example 1.** *Consider* $f(x) = c \cdot |x|^q$, *where* $q > 1$, $c > 0$. $f(x)$ *satisfies Assumption 3 with* $\alpha = \frac{q}{q-1}$ *and* $\mu = \frac{c^{2/q}q^2}{2}$.

**Example 2.** *Consider* $f(x) = \frac{e^x + e^{-x}}{2} - 1$. $f(x)$ *satisfies Assumption 3 with* $\alpha = 1$ *and* $\mu = {}^1\!/2$.

**Example 3.** *Consider* $f(x) = cosh(x) + 8 \cdot cosh(sin(x)) - 9$, *where* $cosh(x) = {}^{(e^x + e^{-x})}\!/2$. *The derivative is* $f'(x) = sinh(x) + 8 \cdot cos(x) \cdot sinh(sin(x))$ *and* $|f'(x)| \geq 10^{-2} \cdot f(x)$ *for all* $x$. *Then* $f(x)$ *satisfies Assumption 3 with* $\alpha = 1$ *and* $\mu = 5 \cdot 10^{-5}$.

Note that the functions in Example 1 and Example 2 are convex, whereas the function in Example 3 is nonconvex.

The following proposition shows that KŁ property is preserved under some operators such as direct addition.

**Proposition 3.** *Let* $f(\cdot)$ *be a separable function, i.e.,* $f(x) := \frac{1}{n}\sum_{i=1}^{n} f_i(x_i)$, *where* $x = (x_1, \ldots, x_n)$, $x_i \in \mathbb{R}^{d_i}$, $\sum_{i=1}^{n} d_i = d$. *Let each* $f_i(\cdot)$ *satisfy KŁ inequality (Assumption 2) with* $\phi_i(t)$. *Then* $f(\cdot)$ *also satisfies KŁ inequality with* $\phi(t) := \frac{1}{\sqrt{n}}\min_{1\leq i\leq n}\phi_i(t)$.

*Proof.* By separability and KŁ condition we have

$$
\begin{aligned}
\|\nabla f(x)\|^2 &= \sum_{i=1}^{n}\frac{1}{n^2}\|\nabla f_i(x_i)\|^2 \\
&\geq \sum_{i=1}^{n}\frac{1}{n^2}\phi_i^2\left(f_i(x_i) - f_i^{inf}\right) \\
&\overset{(i)}{\geq} \frac{1}{n}\sum_{i=1}^{n}\phi^2\left(f_i(x_i) - f_i^{inf}\right) \\
&\overset{(ii)}{\geq} \phi^2\left(\frac{1}{n}\sum_{i=1}^{n}f_i(x_i) - f_i^{inf}\right) \\
&\overset{(iii)}{\geq} \phi^2\left(f(x) - f^{inf}\right),
\end{aligned}
\tag{9}
$$

where $(i)$ holds by definition of $\phi(t)$, $(ii)$ is due to convexity of $\phi(t) := \frac{1}{\sqrt{n}}\min_{1\leq i\leq n}\phi_i(t)$ and Jensen's inequality and $(iii)$ follows from $\frac{1}{n}\sum_{i=1}^{n}\inf_{x_i} f_i(x_i) \leq \inf_x \frac{1}{n}\sum_{i=1}^{n} f_i(x)$ for any $x = (x_1, \ldots, x_n)$. $\qquad\square$

The above Proposition 3 implies, in particular, that if we have a separable function $f(x) = \sum_{i=1}^{n} f_i(x_i)$ and each $f_i(x_i)$ is 1-PŁ with $\mu_i$, $i = 1, \ldots, n$, then $f(x)$ satisfies 1-PŁ with $\mu = \frac{\mu_{min}}{n}$.

**Example 4.** *Consider* $f(x, y) = cosh(x) + 8 \cdot cosh(sin(x)) + 0.5 \cdot cosh(y) + 2.5 \cdot cosh(sin(y)) - 12$. *This function of two variables satisfies Assumption 3 with* $\alpha = 1$ *and* $\mu = 5 \cdot 10^{-5}$.

Now we list several problems which occur in applications and satisfy $\alpha$-PŁ with $\alpha = 1$.

**Example 5** (Policy gradient optimization in RL). *Consider a Markov Decision Process (MDP)* $M = \{\mathcal{S}, \mathcal{A}, \mathcal{P}, \mathcal{R}, \gamma, \rho\}$, *where* $\mathcal{S}$ *is a state space;* $\mathcal{A}$ *is an action space;* $\mathcal{P}$ *is a transition model, where* $\mathcal{P}(s'|s, a)$ *is the transition density to state* $s'$ *from a given state* $s$ *under a given action* $a$; $\mathcal{R} = \mathcal{R}(s, a)$ *is the bounded reward function for state-action pair* $(s, a)$; $\gamma \in [0, 1)$ *is the discount factor; and* $\rho$ *is the initial state distribution. The behavior of the agent in MDP is characterized by the*

*parametric policy $\pi_\theta(a|s)$ over $\mathcal{S} \times \mathcal{A}$, which denotes the probability of taking action $a$ at the state $s$. The policy $\pi_\theta$ is assumed to be differentiable with respect to parameter $\theta \in \mathbb{R}^d$. Let $\tau = \{s_t, a_t\}_{t\geq 0}$ be a trajectory generated by the policy $\pi_\theta$ and it is distributed according to distribution $\tau \sim p(\tau|\pi_\theta)$. The expected return of the policy $\pi_\theta$ is defined by*

$$J(\theta) := \mathbb{E}_\tau \left[ \sum_{t=0}^\infty \gamma^t \mathcal{R}(s_t, a_t) \right].$$

*The goal of policy-based methods is to find $\theta$ which maximizes the expected return $\theta^\star \in \arg\max_\theta J(\theta)$. It was recently shown that the above objective satisfies 1-PŁ assumption*

$$\|\nabla J(\theta)\| \geq \sqrt{2\mu}\left(J^\star - J(\theta)\right) \qquad \text{for all } \theta \in \mathbb{R}^d$$

*under the standard assumptions on $\pi_\theta$ and $\rho$ such as non-degenerate Fisher matrix and transferred compatible function approximation error [41, 1, 56].*

**Example 6** (Operations management problems). *In applications such as supply chain or revenue management [14], problems can often be formulated as*

$$\min_{x \in \mathcal{X}} F(x) := \mathbb{E}\left[\phi(x \wedge \xi)\right], \tag{10}$$

*where $\mathcal{X}$ is a convex compact subset of $\mathbb{R}^d$, $\xi$ is a random vector, $\wedge$ denotes a component-wise minimum and $\phi(\cdot)$ is convex. As a result, $F(\cdot)$ becomes non-convex. On the other hand, such problem often admits a convex reformulation*

$$\min_{y \in \mathcal{Y}} G(y) := F(g^{-1}(y)), \tag{11}$$

*where $g(x) = \mathbb{E}\left[x \wedge \xi\right]$ and function $G(\cdot)$ is convex. Suppose $g : \mathcal{X} \to \mathcal{Y}$ is a bijective differentiable map with $\nabla g(x) \succeq \lambda I$, $\lambda > 0$ for all $x \in \mathcal{X}$, then function $F(\cdot)$ satisfies 1-PŁ condition. This is because: for any $x$ with $g(x) = y$,*

$$
\begin{aligned}
F(x) - F(x^\star) &= G(y) - G(y^\star) \\
&\leq \langle \nabla G(y), y - y^\star \rangle \\
&\leq \|\nabla G(y)\| \|y - y^\star\| \\
&= \|\nabla g^{-1}(y)\nabla F(x)\| \|y - y^\star\| \\
&\leq \frac{D_\mathcal{Y}}{\lambda}\|\nabla F(x)\|,
\end{aligned}
$$

*where $D_\mathcal{Y}$ is the diameter of the set $\mathcal{Y}$. Therefore, $F(\cdot)$ is 1-PŁ with $\mu = \frac{1}{2}\frac{\lambda^2}{D_\mathcal{Y}^2}$.*

**Remark 2.** *Note that even though the problem in Example 6 satisfies 1-PŁ condition, our theory developed in this work is not directly applicable to solve this problem The reason is that this problem has a compact constraint and therefore requires an appropriate generalization of PŁ condition, e.g., using the notion of gradient mapping or the subgradient of the indicator function of the set $\mathcal{X}$, see [29] for examples. However, our theory becomes applicable for this problem if we additionally assume that the solution of (10) lies in the interior of $\mathcal{X}$ and all the iterates $\{x_t\}_{t\geq 0}$ generated by the method remain in the interior of $\mathcal{X}$.*

### A.2  KŁ Functions

**Example 7.** *A commonly used type of loss function in machine learning applications is a squared cross entropy (CE), it is given by*

$$\ell(x, y) := \sum_i y_i \log\left(\frac{e^{x_i}}{\sum_j e^{x_j}}\right)^2.$$

*Under such loss function, it is known [51] that KŁ condition holds with corresponding function $\phi(t) = \min\{t, \sqrt{t}\}$. This is function is both positive and $\phi(t)^2$ is convex. Next, we apply the result of Theorem 1 to obtain the convergence rate of SGD for this type of loss functions assuming the*

*stochastic gradient estimator satisfying Assumption 4 with $h(t) = t$. First step is to obtain the stationary point $r(\eta)$ using Equation (7).*

$$2a\eta^2 t + 2\frac{d\eta^2}{b} = \eta\big(\min\{t, \sqrt{t}\}\big)^2.$$

*It is straightforward to see that for small enough $\eta$, the stationary point is smaller than 1. In this case, $\min\{t, \sqrt{t}\}$ is t. Therefore, we are in the setting of Corollary 1 with $\alpha = 1, \beta = 1$, and $\tau = 0$. This implies that the interation (and sample) complexity of $\mathsf{SGD}$ is of the order $\mathcal{O}(\epsilon_f^{-3})$. Moreover, if A in Assumption 4 is zero, using a similar argument and the result of Theorem 2, one can derive that $\mathsf{PAGER}$ give us $\mathcal{O}(\epsilon_f^{-2})$ sample complexity.*

To illustrate the generality of the result of Theorem 1, next we present the convergence rate of SGD for objective functions that satisfy the global KŁ condition with $\phi(t) = \sqrt{t\log(t+1)}$ under Assumption 4 with $h(t) = \log(1+t)$.

**Example 8.** *Consider the scenario in which the objective function satisfies the global KŁ condition with $\phi(t) = \sqrt{t\log(t+1)}$ and a stochastic gradient estimator satisfies Assumption 4 with $h(t) = \log(1+t)$. In this case, Equation (7) becomes*

$$2a\eta^2 \log(1+t) + 2\frac{d\eta^2}{b} = \eta t \log(1+t).$$

*Defining $u := \log(t+1)$ yields*

$$\eta\big(2au + 2\frac{d}{b}\big) = (e^u - 1)u \approx (u + \frac{u^2}{2})u.$$

*The last approximation is true since for small enough $\eta$, u is less than one. Solving the above cubic equation leads to a solution that is of the order $u = \Theta(\sqrt{\eta})$ or equivalently $r(\eta) = \Theta(\exp(\sqrt{\eta}) - 1)$. Note that for small enough $\eta \ll 1$, we have $\Theta(\exp(\sqrt{\eta}) - 1) = \Theta(\sqrt{\eta} + \eta) = \Theta(\sqrt{\eta})$, i.e., $\nu = 0.5$. To obtain $\zeta$ in Theorem 1, we use Equation (8) which leads to*

$$1 + \frac{a\eta_k^2}{1 + r(\eta_k)} - \frac{\eta_k}{2}\Big(\frac{r(\eta_k)}{1 + r(\eta_k)} + \log(1 + r(\eta_k))\Big)$$

$$= 1 + \frac{a\eta_k^2}{1 + \sqrt{\eta_k}} - \frac{\eta_k}{2}\Big(\frac{\sqrt{\eta_k}}{1 + \sqrt{\eta_k}} + \log(1 + \sqrt{\eta_k})\Big) = 1 - \omega_k k^{-1},$$

*In order to have the above equality, we can have $\eta_k = \Theta(k^{-1})$. Finally, the result of Theorem 1 yields $\delta_k = \mathcal{O}(k^{-\zeta\nu}) = \mathcal{O}(k^{-0.5})$.*

## B    Proofs for Section 3

### B.1    Proof of Lemma 1

**Lemma 1.** Under Assumptions 1, 2, and 4 with constant cost, i.e., $b := b_k$, we obtain

$$\delta_{t+1} \le \delta_t + a\eta^2 \cdot h\big(\delta_t\big) - \frac{\eta}{2}\phi^2(\delta_t) + \frac{d\eta^2}{b},$$

where $\delta_t := \mathbb{E}\left[f(x_t) - f^\star\right]$, $a := LA$, $d := \frac{LC}{2}$, $\eta := \eta_k$.

*Proof.* Let $\{x_0, x_1, x_2, ...\}$ denote the sequence of points that are obtained from SGD. From the L-smoothness assumption, we obtain

$$f(x_{t+1}) \le f(x_t) - \eta\langle\nabla f(x_t), g_k(x_t, \xi_t)\rangle + \frac{L}{2}||x_{t+1} - x_t||^2.$$

Taking the conditional expectation of both side of the above inequality given $x_t$ yields

$$\mathbb{E}[f(x_{t+1}) - f(x_t)|x_t] \le -\eta\mathbb{E}\big[\langle\nabla f(x_t), g_k(x_t, \xi_t)\rangle|x_t\big] + \frac{L}{2}\eta^2\mathbb{E}\big[||g_k(x_t, \xi_t)||^2|x_t\big].$$

Using Assumption 4 and the fact that oracle's queries are unbiased, we obtain

$$\mathbb{E}[f(x_{t+1}) - f(x_t)|x_t] \leq LA\eta^2 \cdot h\big(f(x_t) - f^*\big) - \eta(1 - \frac{L}{2}\eta B) \cdot \phi^2\big(f(x_t) - f^*\big) + \frac{L}{2}\eta^2\frac{C}{b},$$

where $b = b_k$ denotes the cost of gradient $g_k$. Since the choice of learning rate is ours, we select it such that $(1 - \frac{L}{2}\eta B) \geq \frac{1}{2}$. Using Assumption 2 for points around the optimum point $x^*$, we obtain

$$\mathbb{E}[\varrho_{t+1}|x_t] - \varrho_t \leq a\eta^2 \cdot h\big(\varrho_t\big) - \frac{\eta}{2}\phi^2(\varrho_t) + \frac{d\eta^2}{b},$$

where $\varrho_t := f(x_t) - f^*$, $a := LA$, and $d := \frac{LC}{2}$. Let $\delta_t = \mathbb{E}[\varrho_t]$. Using the fact $h(t)$ is concave and $\phi^2$ is convex, and Jensen's inequality, we obtain the result. $\square$

## B.2 Proof of Theorem 1

We first prove the following technical lemma.

**Lemma 2.** *Consider a series $\{r_t\}_{t \geq 0}$ that for every integer $T > 0$ satisfies the following inequality*

$$r_t \leq \prod_{i=1}^{k}(1 - a_i i^{-1})^T r_0 + \mathcal{O}(k^{-b}),$$

*where $t = kT$ and $a < a_i < A \leq 1$ for some positive constants $a$ and $A$ and all $i$. Then, there exists $T$ such that $r_t = \mathcal{O}(t^{-b})$.*

*Proof.* Using the fact that $a_i$s are bounded and $1 - x \leq \exp(-x)$, we obtain

$$\prod_{i=1}^{k}(1 - a_i i^{-1})^T \leq \exp\big(-aT\sum_{i=1}^{k}i^{-1}\big) \leq (k+1)^{-aT}.$$

In the above inequality, we used $\sum_{i=1}^{k} i^{-1} \geq \int_1^{k+1} x^{-1}dx = \log(k+1)$. Selecting $T = \lceil b/a \rceil$ will imply the result. $\square$

**Theorem 1.** Suppose there exist $\nu, \{\omega_j\}_{j \geq 0}$, and $\zeta \geq 0$ such that $\eta_k = \Theta(k^{-\zeta})$, $r(\eta_k) = \Theta(k^{-\zeta\nu})$, $|1 - \omega_k| < 1$, and

$$1 + a\eta_k^2 h'\big(r(\eta_k)\big) - \eta_k\phi'\big(r(\eta_k)\big)\phi\big(r(\eta_k)\big) = 1 - \omega_k k^{-1}.$$

Then, $\delta_k = \mathcal{O}(k^{-\zeta\nu})$ and the iteration complexity of Algorithm 1 is $\mathcal{O}(\epsilon_f^{-1/(\zeta\nu)})$.

*Proof.* Suppose, we are in the $k$ iteration of the outer-loop of Algorithm 1. Using Lemma 1 and the definition of $r(\eta)$ in (7), we have

$$\delta_{t+1} \leq \delta_t + a\eta_k^2\Big(h\big(\delta_t\big) - h\big(r(\eta_k)\big)\Big) - \frac{\eta_k}{2}\Big(\phi^2(\delta_t) - \phi^2\big(r(\eta_k)\big)\Big).$$

By defining $y_t := \delta_t - r(\eta_k)$ and using the concavity of functions $h(\cdot)$ and $-\phi^2(\cdot)$, we obtain

$$y_{t+1} \leq y_t\Big(1 + a\eta_k^2 h'\big(r(\eta_k)\big) - \eta\phi'\big(r(\eta_k)\big)\phi\big(r(\eta_k)\big)\Big).$$

Given the assumption in Theorem 1, we have

$$y_{t+1} \leq y_t\big(1 - \omega_k k^{-1}\big).$$

Recall that $k$ corresponds to the index of the outer-loop. After $t$ iterations of the inner-loop (in which index $k$ is fixed), we obtain

$$y_t \leq y_0\big(1 - \omega_k k^{-1}\big)^t. \tag{12}$$

This shows the rate at which the inner-loop of Algorithm 1 (lines 4-5) converges to point $x$, where $r(\eta_k) = f(x) - f^*$. Based on Equation (12), after setting $\eta = \eta_1$ and $T_1$ rounds of the inner-loop, we obtain

$$y_{T_1} \leq y_0\big(1 - \omega_1\big)^{T_1} \Rightarrow \delta_{T_1} \leq y_0\big(1 - \omega_1\big)^{T_1} + r(\eta_1).$$

Continuing this process, after updating $\eta = \eta_2$ and going through the inner-loop for another $T_2$ iterations imply

$$y_{T_1+T_2} \le (\delta_{T_1} - r(\eta_2))\left(1 - \frac{\omega_2}{2}\right)^{T_2} \Rightarrow \delta_{T_1+T_2} \le \left(y_0(1-\omega_1)^{T_1} + r(\eta_1) - r(\eta_2)\right)\left(1 - \frac{\omega_2}{2}\right)^{T_2} + r(\eta_2).$$

The above inequality is because before starting the inner-loop for the second round, the initial point for $y$ is $y_{T_1} := \delta_{T_1} - r(\eta_2)$. Using induction and after $k$ rounds of outer-loop, we obtain

$$\delta_t \le \prod_{i=1}^{k} \left|1 - \frac{\omega_i}{i}\right|^{T_i} y_0 + \prod_{i=1}^{k}\left|1 - \frac{\omega_i}{i}\right|^{T_i}\left(\sum_{j=1}^{k-1} \frac{r(\eta_j) - r(\eta_{j+1})}{\prod_{i'=1}^{j}\left(1 - \frac{\omega_{i'}}{i'}\right)^{T_{i'}}}\right) + r(\eta_k)$$

$$= \prod_{i=1}^{k}\left|1 - \frac{\omega_i}{i}\right|^{T_i} y_0 + \sum_{j=1}^{k-1} \left(r(\eta_j) - r(\eta_{j+1})\right) \prod_{i=j+1}^{k}\left|1 - \frac{\omega_i}{i}\right|^{T_i} + r(\eta_k)$$

where $t = \sum_j T_j$. In Algorithm 1, $T_i$ are selected to be $T$. Next, using Lemma 2, we show there exist a positive constant $T$ such that $\delta_t = \mathcal{O}(t^{-\nu\zeta})$. Following the proof of Lemma 2, we have

$$\delta_t \le \prod_{i=1}^{k}\left|1 - \frac{\omega_i}{i}\right|^{T} y_0 + \sum_{j=1}^{k-1}\left(r(\eta_j) - r(\eta_{j+1})\right) \prod_{i=j+1}^{k}\left|1 - \frac{\omega_i}{i}\right|^{T} + r(\eta_k)$$

$$\le (k+1)^{-\omega T} y_0 + \sum_{j=1}^{k-1}\left(r(\eta_j) - r(\eta_{j+1})\right)\left(\frac{j+1}{k+1}\right)^{\omega T} + r(\eta_k),$$

where $t = kT$ and $\omega = \min_i \omega_i$. Let $b := \nu\zeta$ and $T := \lceil (b+1)/\omega \rceil$. Since $r(\eta) = \Theta(\eta^\nu)$ and $\eta_j = \Theta(j^{-\zeta})$, then there exists a constant $C > 0$ such that

$$\delta_t \le (k+1)^{-\omega T} y_0 + C\sum_{j=1}^{k-1}\left(j^{-b} - (j+1)^{-b}\right)\left(\frac{j+1}{k+1}\right)^{\omega T} + \mathcal{O}(k^{-b})$$

$$\le \mathcal{O}(k^{-b}) + \frac{C}{(k+1)^{b+1}}\sum_{j=1}^{k-1}\left((1 + \frac{1}{j})^b - 1\right)(j+1) + \mathcal{O}(k^{-b}).$$

Using $(1+x)^b - 1 \le bx/(1-bx)$ for $x < 1/b$ and $b \ge 0$, we obtain

$$\delta_t \le \mathcal{O}(k^{-b}) + \frac{C'}{(k+1)^{b+1}} + \frac{Cb}{(k+1)^{b+1}}\sum_{j>b}^{k-1} \frac{j+1}{j-b} + \mathcal{O}(k^{-b}) = \mathcal{O}(k^{-b}).$$

where $C' \ge 0$ is a constant corresponding to the part of the summation for $j \le b$. The result follows from the fact that $k = t/T$ and $T$ is a constant. $\qquad\square$

### B.3   Proof of Corollary 1

**Corollary 1.** Consider a special case of Assumption 4 with $h(t) = t^\beta$ and $b_k = k^\tau$, where $\beta \in (0,1]$ and $\tau \ge 0$. Suppose the objective function $f$ satisfies Assumptions 1 and 3. Let $\gamma := \alpha\beta$. Then, for any $\epsilon_f > 0$, Algorithm 1 returns a point $x$ with $\mathbb{E}\left[f(x) - f^\star\right] \le \epsilon_f$ after $N := K \cdot T$ iterations.
i) When $\gamma = 2$ ($\alpha = 2$ and $\beta = 1$), we have

$$N = \mathcal{O}(\epsilon_f^{-\frac{1}{1+\tau}}), \text{ with } \eta_k = \Theta(k^{-1}).$$

ii) When $\gamma < 2$, we have

$$N = \mathcal{O}\big(\epsilon_f^{-\frac{4-\alpha}{\alpha(\tau+1)}}\big) \text{ with } \eta_k = \Theta(k^{-\frac{\tau+1}{2-\alpha/2}+\tau}) \text{ if } \tau \le \frac{\gamma}{4-\alpha-\gamma}, and$$

$$N = \mathcal{O}\big(\epsilon_f^{-\frac{4-\alpha-\gamma}{\alpha}}\big) \text{ with } \eta_k = \Theta(k^{-\frac{2-\gamma}{4-\alpha-\gamma}}) \text{ if } \tau > \frac{\gamma}{4-\alpha-\gamma}.$$

*Proof.* Using the result of Theorem 1, we need to specify the constants $\nu$ and $\zeta$. To do so, we first characterize the stationary point for the special setting of this corollary. Equation (7) becomes

$$a\eta \cdot t^\beta + \frac{d\eta}{b} = \mu t^{2/\alpha}. \tag{13}$$

Let $\gamma := \alpha\beta$ and define the following function

$$H_\eta(t) := a\eta^2 t^\beta - \mu\eta t^{\frac{2}{\alpha}} + \frac{d\eta^2}{b}.$$

Next, we either find $r(\eta)$ exactly or bound it. Depending on whether $\gamma$ is less than or equal to 2, the analysis of $H_\eta(t) = 0$ is different. We study each case separately.

**I)** $\gamma = 2$ (or $\beta = 1$ and $\alpha = 2$): In this case, we can find $r(\eta)$ exactly and it is given by

$$r(\eta) = \frac{\frac{d\eta}{b}}{\mu - a\eta} = \Theta\left(\frac{\eta}{b}\right) = \Theta\left(k^{-\tau}\eta\right).$$

Note that in the above expression, we used the fact that $b_k = \Theta(k^\tau)$. Next is to find the parameters in Theorem 1. To do so, from Equation 8 with $h(t) = t$ and $\phi(t) = \sqrt{2\mu t}$, we have

$$1 + a\eta_k^2 h'\left(r(\eta_k)\right) - \eta_k \phi'\left(r(\eta_k)\right)\phi\left(r(\eta_k)\right) = 1 + a\left(\Theta(k^{-\zeta})\right)^2 - \mu\Theta(k^{-\zeta}) = 1 - \omega_k k^{-1}.$$

In order to have the above equality, we should have $\zeta = 1$. Now, suppose that $\tau \geq 0$, then $r(\eta) = \Theta(k^{-(1+\tau)})$ and based on Theorem 1, we obtain the convergence rate of $\mathcal{O}(k^{-(1+\tau)})$ for $\delta_k$.

**II)** $0 \leq \gamma < 2$: In this case, we find lower and upper bound for $r(\eta)$. To this end, consider the following point for some constant $S$,

$$t_0 := \left(\frac{\frac{d\eta}{b} + S\frac{\eta}{b}}{\mu}\right)^{\frac{\alpha}{2}} = \Theta\left(\left(\frac{\eta}{b}\right)^{\frac{\alpha}{2}}\right).$$

For this point, we have

$$H_\eta(t_0) = a\frac{\eta^{2+\frac{\gamma}{2}}}{b^{\frac{\gamma}{2}}}\left(\frac{d+S}{\mu}\right)^{\frac{\gamma}{2}} - S\frac{\eta^2}{b}.$$

For $S = 0$, $H_\eta(t_0) > 0$. On the other hand, if $\eta = \Theta(k^{-\zeta})$ and $b_k = \Theta(k^\tau)$, then for $(\tau + \zeta)\frac{\gamma}{2} \geq \tau$ and large enough $S$, we have $H_\eta(t_0) < 0$. This implies

$$r(\eta) = \Theta\left(\left(\frac{\eta}{b}\right)^{\frac{\alpha}{2}}\right).$$

Next is to check whether (8) holds for $\eta_k = \Theta(k^{-\zeta})$, $b_k = \Theta(k^\tau)$, $h(t) = t^\beta$, and $\phi(t) = \sqrt{2\mu}t^{2/\alpha}$, i.e.,

$$1 + a\beta\left(\Theta(k^{-\zeta})\right)^2\left(\Theta(k^{-(\zeta+\tau)\alpha/2})\right)^{\beta-1} - \frac{2\mu}{\alpha}\Theta(k^{-\zeta})\left(\Theta(k^{-(\zeta+\tau)\alpha/2})\right)^{2/\alpha-1} = 1 - \omega_k k^{-1}.$$

The order of the first term is $\mathcal{O}(k^{-(2\zeta+(\zeta+\tau)\alpha(\beta-1)/2)})$ and the order of the second term is $\mathcal{O}(k^{-(\zeta+(\zeta+\tau)\alpha(2/\alpha-1)/2)})$. In order for the above expression to hold, we should have

$$\zeta + (\zeta + \tau)\alpha(2/\alpha - 1)/2 \leq 2\zeta + (\zeta + \tau)\alpha(\beta - 1)/2, \tag{14}$$

and

$$\zeta + (\zeta + \tau)\alpha(2/\alpha - 1)/2 \leq 1. \tag{15}$$

Inequality (14) implies $\gamma\zeta \geq (2 - \gamma)\tau$ and inequality (15) leads to $\zeta < \frac{\tau+1}{2-\alpha/2} - \tau$. See Figure 2 for an example of the region $(\zeta, \tau)$ for which both (14) and (15) hold. Putting everything together, we obtain

$$\text{If } \tau \leq \frac{\gamma}{4 - \alpha - \gamma}, \text{ then } \delta_k = \mathcal{O}(k^{-\frac{\alpha(\tau+1)}{4-\alpha}}), \quad \text{with} \quad \eta_k = \Theta(k^{-\frac{\tau+1}{2-\alpha/2}+\tau}).$$

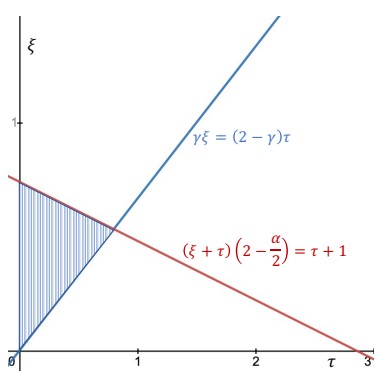

Figure 2: An illustration of the region $(\tau, \zeta)$ that ensures both (14) and (15) hold. This is the highlighted area. Within this region, the maximum $\gamma$ is at the red line. In this figure $\gamma = 1.2, \alpha = 1.3$, and $\beta = 1.2/1.3 = 0.92$.

Note that $\frac{\gamma}{4-\alpha-\gamma}$ is the intersection point of two lines. For $\tau > \frac{\gamma}{4-\alpha-\gamma}$, the dynamic is equivalent to

$$\delta_{t+1} \le \delta_t + a\eta^2 \cdot \delta_t^{\beta} - \eta\mu\delta_t^{\frac{2}{\alpha}}, \tag{16}$$

with the stationary point $r(\eta) = (a\eta/\mu)^{\frac{\alpha}{2-\gamma}}$. Following the steps similar to the previous case, we get the following equation

$$1 + a\beta\big(\Theta(k^{-\zeta})\big)^2 \Big(\Theta(k^{-\zeta\frac{\alpha}{2-\gamma}})\Big)^{\beta-1} - \frac{2\mu}{\alpha}\Theta(k^{-\zeta})\Big(\Theta(k^{-\zeta\frac{\alpha}{2-\gamma}})\Big)^{2/\alpha-1} = 1 - \omega_k k^{-1}.$$

This leads to $\zeta = \frac{2-\gamma}{4-\alpha-\gamma}$ and subsequently to

$$\text{If } \tau > \frac{\gamma}{4-\alpha-\gamma}, \quad \text{then } \delta_k = \mathcal{O}(k^{-\frac{\alpha}{4-\alpha-\gamma}}), \quad \text{with} \quad \eta_k = \Theta(k^{-\frac{2-\gamma}{4-\alpha-\gamma}}).$$

$\square$

### B.4 Proof of Proposition 1

**Proposition 1.** Let the assumptions of Corollary 1 hold, $b_k = \Theta(k^\tau), T = \Theta(1)$. Then the expected total computational cost (sample complexity) of Algorithm 1 is

$$\text{cost} := T \cdot \sum_{k=0}^{K-1} b_k = \begin{cases} \mathcal{O}\left(\epsilon_f^{-\frac{4-\alpha}{\alpha}}\right) & \text{for } 0 \le \tau \le \frac{\gamma}{4-\alpha-\gamma}, \\ \mathcal{O}\left(\epsilon_f^{-\frac{(4-\alpha-\gamma)(\tau+1)}{\alpha}}\right) & \text{for } \tau > \frac{\gamma}{4-\alpha-\gamma}. \end{cases}$$

*Proof.* Corollary 1 says that Algorithm 1 finds the global $\epsilon_f$-stationary point after $N = K \cdot T$ number of iterations, where

$$N = \mathcal{O}\big(\epsilon_f^{-\frac{4-\alpha}{\alpha(\tau+1)}}\big) \text{ with } \eta_k = \Theta(k^{-\frac{\tau+1}{2-\alpha/2}+\tau}) \text{ if } \tau \le \frac{\gamma}{4-\alpha-\gamma},$$

$$N = \mathcal{O}\big(\epsilon_f^{-\frac{4-\alpha-\gamma}{\alpha}}\big) \text{ with } \eta_k = \Theta(k^{-\frac{2-\gamma}{4-\alpha-\gamma}}) \text{ if } \tau > \frac{\gamma}{4-\alpha-\gamma}.$$

For $\tau \le \frac{\gamma}{4-\alpha-\gamma}$, the expected total computational cost (sample complexity) is

$$\text{cost} := T \cdot \sum_{k=0}^{K-1} b_k = T \sum_{k=0}^{K-1} k^\tau = \mathcal{O}(N^{\tau+1}) = \mathcal{O}\left(\epsilon_f^{-\frac{4-\alpha}{\alpha(\tau+1)}\cdot(\tau+1)}\right) = \mathcal{O}\left(\epsilon_f^{-\frac{4-\alpha}{\alpha}}\right).$$

For $\tau > \frac{\gamma}{4-\alpha-\gamma}$, the iteration complexity does not improve and the sample complexity becomes worse when increasing $\tau$

$$\text{cost} = T \cdot \sum_{k=0}^{K-1} b_k = T \sum_{k=0}^{K-1} k^\tau = \mathcal{O}(N^{\tau+1}) = \mathcal{O}\left(\epsilon_f^{-\frac{(4-\alpha-\gamma)(\tau+1)}{\alpha}}\right).$$

$\square$

## B.5 Proof of Proposition 2

**Proposition 2.** Consider the following recursion

$$\delta_{k+1} = \delta_k + a\eta_k^2 \cdot h(\delta_k) - \frac{\eta_k}{2}\phi^2(\delta_k) + \frac{d\eta_k^2}{b_k}, \quad \text{for all } k \geq 0,$$

where $a \geq 0$, $d > 0$, $h(t) = t^\beta$ with $\beta \in (0,1]$, $\phi(t) = \sqrt{2\mu}t^{1/\alpha}$ with $\alpha \in [1,2]$, and $b_k = \Theta(1)$. Then $\delta_k = \Omega(k^{-\frac{\alpha}{4-\alpha}})$ for any sequence of $\{\eta_k\}_{k \geq 0}$. Moreover, this rate is achieved by the choice $\eta_k = \Theta(k^{-\frac{1}{2-\alpha/2}})$.

*Proof.* We begin with the fact that if $\delta_k$ defined in (6) converges to zero with stepsizes $\{\eta_k\}$, then there exists a $K_0$ such that for all $k \geq K_0$, $\delta_k < 1$. Hence, for $k \geq K_0$, we have $\delta_k \leq \delta_k^\beta$ for $\beta \in (0,1]$. An immediate consequence of this fact is that for $h(t) = t^\beta$ and $\phi(t) = \sqrt{2\mu}t^{1/\alpha}$, the above dynamic can be bounded as follows

$$\delta_k + a\eta_k^2\delta_k - \eta_k\mu\delta_k^{\frac{2}{\alpha}} + d\eta_k^2 \leq \delta_k + a\eta_k^2\delta_k^\beta - \eta_k\mu\delta_k^{\frac{2}{\alpha}} + d\eta_k^2, \quad \forall k \geq K_0. \tag{17}$$

Let us define two new dynamics as follows, i.e.,

$$r_{k+1} := r_k + a\eta_k^2 r_k - \eta_k\mu r_k^{\frac{2}{\alpha}} + d\eta_k^2, \quad r_0 := \delta_0, \tag{18}$$

$$r_{k+1,\varepsilon} := r_k\left(1 - a'\eta_k^{1+\frac{2-\alpha-\varepsilon}{2}}\right) + d'\eta_k^2, \quad r_{0,\varepsilon} := \delta_0, \tag{19}$$

First, we show that for any $0 < \varepsilon < \frac{2-\alpha}{2}$, there exist $K, a', d'$, such that for all $k \geq K$, $r_{k+1,\varepsilon} \leq r_{k+1}$. To do so, we need to understand for what values of $z$, the following inequality holds.

$$z\left(1 - a'\eta_k^{1+\frac{2-\alpha-\varepsilon}{2}}\right) + d'\eta_k^2 \leq z + a\eta_k^2 z - \eta_k\mu z^{\frac{2}{\alpha}} + d\eta_k^2.$$

This implies

$$0 \leq \left(a'\eta_k^{1+\frac{2-\alpha-\varepsilon}{2}} + a\eta_k^2\right)z - \eta_k\mu z^{\frac{2}{\alpha}} + (d-d')\eta_k^2. \tag{20}$$

By choosing $d' = d$, the above inequality holds for

$$0 \leq z \leq \left(\frac{a'\eta_k^{\frac{2-\alpha-\varepsilon}{2}} + a\eta_k}{\mu}\right)^{\frac{\alpha}{2}}.$$

Since $a \geq 0$, (20) also holds for

$$z \in \left[0, \left(\frac{a'\eta_k^{\frac{2-\alpha-\varepsilon}{2}}}{\mu}\right)^{\frac{\alpha}{2}}\right].$$

Therefore, if $r_k$ is within the above interval, then $r_{k+1,\varepsilon} \leq r_{k+1}$. Using (17), we know that $r_{k+1} \leq \delta_{k+1}$. On the other hand, based on the result of Theorem 1, we have $\delta_k = \mathcal{O}(\eta_k^{\frac{\alpha}{2}})$. Because of $\frac{2-\alpha-\varepsilon}{2} \leq 1$ and the fact that there exists $K$ such that for all $k \geq K$, $\eta_k \leq 1$, then $\delta_k$ will lay inside the above interval for large enough $k$. This implies that there exists $K'$ such that for all $k \geq K'$, $r_{k+1,\varepsilon} \leq r_{k+1} \leq \delta_{k+1}$. Finally, using the result of Lemma 3 with $\epsilon' = \frac{2-\alpha-\varepsilon}{2}$, we obtain the optimal convergence rate of $r_{k,\varepsilon}$ that is

$$\Theta\left(k^{-\frac{1-\epsilon'}{1+\epsilon'}}\right) = \Theta\left(k^{-\frac{\alpha-\varepsilon}{4-\alpha-\varepsilon}}\right).$$

Comparing the above rate with the rate of $\delta_k$ presented in Corollary 1, i.e., $\mathcal{O}(k^{-\frac{\alpha}{4-\alpha}})$, concludes the result.

$\square$

Next, we present a generalization of Theorem 3.2 in [24] that helps us to establish our tightness result.

**Lemma 3.** *Consider the following recursive equation*

$$r_{k+1} := (1 - a'\eta_k^{1+\epsilon'})r_k + c'\eta_k^2, \ k \geq 0, \tag{21}$$

*where $\eta_k \leq \frac{1}{b'}$ for all $k$ and $a', c', \epsilon' \geq 0$ with $a' \leq b'$. Then, choosing $s \geq 2$ and*

$$\eta_k := \begin{cases} \left(\frac{1}{b'}\right)^{\frac{1}{1+\epsilon'}}, & k < [\frac{K}{2}] \text{ or } K \leq \frac{b'^{\frac{1-\epsilon'}{1+\epsilon'}}}{a'}, \\ \left(\frac{2/(1+\epsilon')}{a'(s+k-[\frac{K}{2}])}\right)^{\frac{1}{1+\epsilon'}}, & \text{otherwise}, \end{cases}$$

*will result in $r_K = \Theta\left(K^{-\frac{1-\epsilon'}{1+\epsilon'}}\right)$.*

*Proof.* For $k \leq [\frac{K}{2}]$, we obtain

$$r_k \leq \left(1 - \frac{a'}{b'}\right)^k r_0 + \frac{c}{b^{\frac{2}{1+\epsilon'}}} \sum_{t=0}^{k-1}(1 - \frac{a'}{b'})^t \leq \left(1 - \frac{a'}{b'}\right)^k r_0 + d_1,$$

where $d_1 := \frac{c'}{a'b'^{\frac{1-\epsilon'}{1+\epsilon'}}}$. Note that if $K \leq \frac{b'^{\frac{1-\epsilon'}{1+\epsilon'}}}{a'}$, then

$$r_K \leq \left(1 - \frac{a'}{b'}\right)^K r_0 + \frac{c'}{a'^2 K},$$

But for $K > \frac{b'^{\frac{1-\epsilon'}{1+\epsilon'}}}{a'}$ and $k = [K/2]$, we have

$$r_{[\frac{K}{2}]} \leq \left(1 - \frac{a'}{b'}\right)^{[\frac{K}{2}]} r_0 + d_1,$$

Then for $k \geq 1 + [\frac{K}{2}]$, we have

$$r_k \leq \left(1 - \frac{2/(1+\epsilon')}{s+k-1-[\frac{K}{2}]}\right) r_{k-1} + c'\left(\frac{2/(1+\epsilon')}{a'(s+k-1-[\frac{K}{2}])}\right)^{\frac{2}{1+\epsilon'}}$$

Multiplying both sides by $e_k := (s+k-1-[\frac{K}{2}])^{\frac{2}{1+\epsilon'}}$ results in

$$e_k r_k \leq \left(s+k-\frac{3+\epsilon'}{1+\epsilon'}-[\frac{K}{2}]\right)\left(s+k-1-[\frac{K}{2}]\right)^{\frac{1-\epsilon'}{1+\epsilon'}} r_{k-1} + c\left(\frac{2}{a'(1+\epsilon')}\right)^{\frac{2}{1+\epsilon'}}$$

$$\leq e_{k-1} r_{k-1} + d_2, \tag{22}$$

where $d_2 := c'\left(\frac{2}{a'(1+\epsilon')}\right)^{\frac{2}{1+\epsilon'}}$. The last inequality is due to the Jensen's inequality and the fact that $\log(x)$ is concave, hence,

$$\left(x - \frac{2}{1+\epsilon}\right)^{1+\epsilon} x^{1-\epsilon} \leq (x-1)^2.$$

Summing up (22) from $k = [K/2]+1$ to $k = K$ gives us

$$e_K r_K \leq e_{[K/2]} r_{[K/2]} + d_2(K - [K/2]).$$

Consequently,

$$r_K \leq \frac{e_{[K/2]}}{e_K} r_{[K/2]} + d_2 \frac{(K-[K/2])}{e_K} = \frac{(s-1)^{\frac{2}{1+\epsilon'}}}{e_K} r_{[K/2]} + d_2 \frac{(K-[K/2])}{e_K}$$

$$\leq \frac{(s-1)^{\frac{2}{1+\epsilon'}}}{e_K}\left(\left(1 - \frac{a'}{b'}\right)^{[\frac{K}{2}]} r_0 + d_1\right) + d_2 \frac{(K-[K/2])}{e_K}.$$

On the other hand, we have $e_K \geq (K - [K/2])^{\frac{2}{1+\epsilon}} \geq (K/2)^{\frac{2}{1+\epsilon}}$, which leads to the following upper bound for $r_K$

$$r_K \leq \frac{(s-1)^{\frac{2}{1+\epsilon'}}}{(K - [K/2])^{\frac{2}{1+\epsilon'}}} \left( \left(1 - \frac{a'}{b'}\right)^{\left[\frac{K}{2}\right]} r_0 + d_1 \right) + \frac{d_2}{(K - [K/2])^{\frac{1-\epsilon'}{1+\epsilon'}}}$$

$$\leq \frac{(s-1)^{\frac{2}{1+\epsilon'}}}{(K/2)^{\frac{2}{1+\epsilon'}}} \left( \left(1 - \frac{a'}{b'}\right)^{\left[\frac{K}{2}\right]} r_0 + d_1 \right) + \frac{d_2}{(K/2)^{\frac{1-\epsilon'}{1+\epsilon'}}}.$$

For the lower bound, we use the following inequality

$$\left(x - \frac{2}{1+\epsilon}\right)^{1+\epsilon} x^{1-\epsilon} \geq (x-2)^2, \quad \forall x \geq 2.$$

This implies

$$e_k r_k \geq e_{k-2} r_{k-1} + d_2.$$

Multiplying the above by $e_{k-1}$, we get

$$e_{k-1} e_k r_k \geq e_{k-1} e_{k-2} r_{k-1} + d_2 e_{k-1}.$$

Summing up the above expression from $k = [K/2] + 1$ to $k = K$ gives us

$$e_{K-1} e_K r_K \geq e_{[K/2]} e_{[K/2]-1} r_{[K/2]} + d_2 \Big( e_{[K/2]} + ... + e_K \Big).$$

Using $\sum_{i=s-1}^{s+K} i^{\frac{2}{1+\epsilon'}} \geq \int_{s-1}^{s+K} x^{\frac{2}{1+\epsilon'}} dx$, we obtain

$$r_K = \Omega\Big( \frac{K^{1 + \frac{2}{1+\epsilon'}}}{K^{\frac{2}{1+\epsilon'}} K^{\frac{2}{1+\epsilon'}}} \Big) = \Omega(K^{\frac{1-\epsilon'}{1+\epsilon'}}).$$

To show that no other designs of stepsizes can achieve better rate, we show that even with the optimal stepsizes, the rate will be the same as above. Note that the dynamic in (21) is a nonlinear function of the stepsize $\eta_k$ that has a global minimum which can be obtained by taking a derivative of (21) with respect to $\eta_k$. This optimal stepsize is given by

$$\eta_k = \left( \frac{a'(1+\epsilon) r_k}{2c'} \right)^{1/(1-\epsilon)}. \tag{23}$$

Using this stepsizes will lead to the following dynamic

$$r_{k+1} = r_k (1 - A r_k^{\frac{2}{1-\epsilon} - 1}), \tag{24}$$

where $A := c'(\frac{1-\epsilon}{1+\epsilon})(\frac{a'(1+\epsilon)}{2c'})^{\frac{2}{1-\epsilon}}$. Given the result of Lemma 6, the convergence rate of this dynamic is $\mathcal{O}(k^{-\frac{1-\epsilon}{1+\epsilon}})$. See Figure 3 for an illustration of an example that shows both the simulated $r_k$ in (24) and its corresponding optimal rate. Different colours show different $\epsilon$.

$\square$

## C  Proofs for Section 4 and Additional Discussion

This Section is organized as follows. First, we elaborate on the intuition why one needs to resort to variance reduction techniques in order to improve over SGD analysis provided in Section 3. Then we highlight the key challenges associated with the analysis of variance reduced methods under global KŁ condition and introduce a new variance reduced method PAGER. We explain the intuition why PAGER overcomes the aforementioned challenges and improves over SGD in online case (1), and over SGD and GD in finite sum (2) case. Finally, we provide convergence guarantees for each setting in Theorems 4 and 5.[11]

---

[11]Note that Theorems 4 and 5 are detailed versions of Theorems 2 and 3 provided in Section 4.

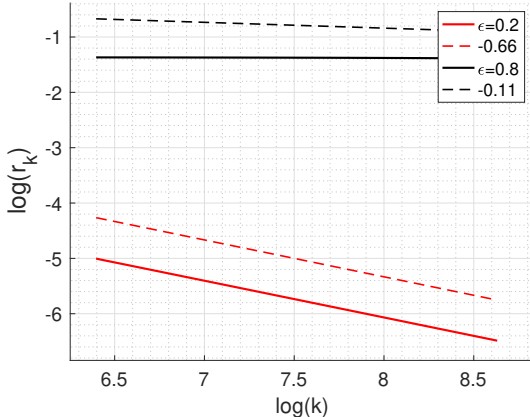

Figure 3: An example to verify equation (24) for $\epsilon \in \{0.2, 0.8\}$. Solid and dashed lines denote the simulated dynamic in Lemma 3 and its corresponding theoretical rates, i.e., $\mathcal{O}(k^{-\frac{1-\epsilon}{1+\epsilon}})$, respectively. Numbers assigned to dashed lines indicate the slope of those lines.

**Why SGD is not enough?**  Notice that the analysis in Section 3, in particular, implies that if we want to solve problem (1) using SGD with constant step-size $\eta$ and a mini-batch with replacement gradient estimator of size $b$, we immediately obtain a recursion

$$\delta_{t+1} - \delta_t \leq -\eta\mu\delta_t^{\frac{2}{\alpha}} + \frac{\eta^2 L\sigma^2}{2b}. \tag{25}$$

It is easy to see that if $\eta$ is fixed, then the last (variance) term in the above recursion can be only controlled by selecting large enough $b$.[12] Assume that we want to solve our problem to $\epsilon_f$ accuracy ($\delta_T \leq \epsilon_f$). Then to balance the two terms on the RHS, one needs to take $b \sim \epsilon_f^{-\frac{2}{\alpha}}$. This choice simplifies the recursion to $\delta_{t+1} - \delta_t \leq -\frac{\eta\mu}{2}\delta_t^{\frac{2}{\alpha}}$. Applying Lemma 6 with $c = \frac{2-\alpha}{\alpha}$, we conclude that one needs $T \sim \epsilon_f^{\frac{-(2-\alpha)}{\alpha}}$ iterations to reach $\delta_T \leq \epsilon_f$. Thus, the total sample complexity is $b \cdot T \sim \epsilon_f^{\frac{-(4-\alpha)}{\alpha}}$. This observation implies that we need to construct a more sophisticated gradient estimator than mini-batch estimator in order to improve the sample complexity of SGD.

**Variance reduction and challenges under KŁ condition.**  One common technique to design faster algorithms in stochastic optimization is to reduce variance of the gradient estimator using a control variate. It turns out that using such variance reduction techniques one can often design a gradient estimator at a much lower cost, while maintaining the same iteration complexity. Let us turn our attention to one popular variance reduction mechanism called PAGE. The main steps of PAGE method is described in Section 4, the detailed pseudo-code is presented in Algorithm 3. This method was originally proposed and analyzed for general non-convex and 2-PŁ objectives [36]. However, its application to $\alpha$-PŁ functions with $\alpha \in [1, 2)$ remains elusive. If we try to apply the standard analysis of PAGE, it will become apparent that we face several challenges. In particular, Lemma 8 along with Lemma 4 provides the following inequality for the iterates of the Algorithm 3

$$\Psi_{t+1} - \Psi_t \quad \leq \quad -\eta\mu\Psi_t^{\frac{2}{\alpha}} - \frac{p_t\lambda_t}{2}G_t\left(1 - \frac{4\eta\mu}{p_t\alpha}\Psi_t^{\frac{2-\alpha}{\alpha}}\right) + \frac{p_t\lambda_t}{2}\frac{\sigma^2}{b_t}, \tag{26}$$

where $\Psi_t = \delta_t + \lambda G_t$ is a candidate for a Lyapunov function and $G_t$ is the variance of the gradient estimator, and $\lambda > 0$. To illustrate one key obstacle in the analysis of PAGE in online setting, let us set $G_t = 0$ for simplicity

$$\Psi_{t+1} - \Psi_t \leq -\eta\mu\Psi_t^{\frac{2}{\alpha}} + \frac{p\lambda\sigma^2}{2b}. \tag{27}$$

---

[12]The results of Corollary 1 and Lemma 1 implies that changing $\eta$ and $b$ with iterations does not help.

Now, this recursion is very similar to (25). Therefore, the same argument applies here. In particular, one can argue that given constant parameters $\eta$, $b'$ and $p$, we need to take $b \sim \epsilon_f^{-\frac{2}{\alpha}}$. Thus the total sample complexity is again no better than $b \cdot T \sim \epsilon_f^{\frac{-(4-\alpha)}{\alpha}}$. Note that the assumption $G_t = 0$ was only made to illustrate one difficulty. Rigorously proving the fact that the term including $G_t$ is small constitutes another challenge.

**Faster rates via PAGER in online case.** However, we notice that in (26), we have one more degree of freedom – the parameter $p$, which can be selected small enough to ensure smaller per iteration cost of the method. This intuition brings us to PAGER (Algorithm 2), a new modification of PAGE method with varying parameter $p$. [13] We carefully select the sequences $\{p_k\}_{k\geq 0}$, $\{b_k\}_{k\geq 0}$, $\{b'_k\}_{k\geq 0}$ for PAGER in order to obtain a small per iteration cost of order $p_k b_k + b'_k \sim \epsilon_f^{-1}$. This leads to a much faster convergence with $\epsilon_f^{-2/\alpha}$ sample complexity.

**Difficulties in finite sum case and a fix via PAGER framework.** Let us now consider a finite sum problem (2) and directly apply Algorithm 3 with (constant) parameters $\eta$, $p$, $b$, $b'$. Then we arrive at the following recursion

$$
\begin{aligned}
\Psi_{t+1} - \Psi_t &\leq -\eta\mu\left(\Psi_t - \lambda G_t\right)^{\frac{2}{\alpha}} - \frac{p\lambda}{2}G_t \\
&\leq -\eta\mu\Psi_t^{\frac{2}{\alpha}} - \frac{p\lambda}{2}G_t\left(1 - \frac{4\eta\mu(n+1)}{\alpha}\Psi_t^{\frac{2-\alpha}{\alpha}}\right),
\end{aligned}
$$

where we applied Lemma 8, 4 and selected optimal parameters $p = \frac{1}{n+1}$, $b = n$, $b' = 1$. By choosing a small enough stepsize $\eta$, we can unroll the above recursion and obtain the sample complexity $\mathcal{O}\left((n\delta_0 + \sqrt{n}\kappa)\left(\frac{1+\delta_0}{\epsilon_f}\right)^{\frac{2-\alpha}{\alpha}}\right)$, where $\delta_0 = f(x_0) - f^\star$, $\kappa = \mathcal{L}/\mu$. However, this complexity is clearly not what one should hope for when analyzing a variance reduction scheme for problem (2). Notably, this complexity can be even worse than the one of standard GD, which is $\mathcal{O}\left(n\kappa\left(\frac{1+\delta_0}{\epsilon_f}\right)^{\frac{2-\alpha}{\alpha}}\right)$, for instance, when $\delta_0 > \kappa$. The main reason for this slowdown is that in the analysis of Algorithm 3 with constant parameters, we are forced to take small step-sizes of order $\eta = \mathcal{O}\left(\frac{1}{n\delta_0}\right)$ to ensure progress. Luckily, thanks to a flexible choice of parameters in PAGER, we can overcome this difficulty and provide improved convergence guaranties. Specifically, the framework of Algorithm 2 allows us to select an increasing sequence of step-sizes until it reaches the value $\eta = \mathcal{O}\left(\frac{1}{\sqrt{n}\mathcal{L}}\right)$.

---

**Algorithm 3: PAGE**

---

1: Initialization: $x_0, g_0 \in \mathbb{R}^d$, step-size $\eta$ , number of iterations $T$, probability $p$, batch-sizes $b$, $b'$
2: **for** $t = 0, \ldots, T-1$ **do**
3:     $x_{t+1} = x_t - \eta g_t$
4:     Sample $\chi \sim \text{Bernoulli}(p)$
5:     **if** $\chi = 1$ **then**
6:         $g_{t+1} = \frac{1}{b}\sum_{i=1}^{b} \nabla f_{\xi_{t+1}^i}(x_{t+1})$
7:     **else**
8:         $g_{t+1} = g_t + \frac{1}{b'}\sum_{i=1}^{b'} \nabla f_{\xi_{t+1}^i}(x_{t+1}) - \frac{1}{b'}\sum_{i=1}^{b'} \nabla f_{\xi_{t+1}^i}(x_t)$
9: **Return:** $x_T$

---

### C.1    Proof of Theorem 2

Now we state and prove a detailed version of Theorem 2.

---

[13]Note that originally PAGE was only analyzed with constant parameter $p$, the extension to an arbitrarily changing $p_t$ is not trivial.

**Theorem 4.** *Let $f(\cdot)$ have the form (1) and satisfy Assumptions 1, 3 (with $\alpha \in [1, 2)$), 5 and 6, let the sequences in Algorithm 2 be chosen as*

$$b_k' = \frac{\alpha}{8\eta\mu}\left(\frac{2^k}{\bar{\Psi}_0}\right)^{\frac{2-\alpha}{\alpha}}, \quad p_k = \frac{1}{1+b_k'},$$

$$b_k = \left(\frac{2 \cdot 2^{\frac{2-\alpha}{\alpha}} \cdot 2^k}{\bar{\Psi}_0}\right)^{\frac{2}{\alpha}} \frac{\sigma^2}{4\mu\eta^2\mathcal{L}^2},$$

$$T_k = \frac{2}{\eta\mu}\left(2 \cdot 2^{\frac{2-\alpha}{\alpha}}\left(\frac{2^k U}{\bar{\Psi}_0} + 2\left(\frac{\eta\mu}{2}\right)^{\frac{\alpha}{2-\alpha}}\right)\right)^{\frac{2-\alpha}{\alpha}},$$

$$\eta_k = \eta = \frac{1}{\mu}\min\left\{\frac{1}{2\kappa}, \frac{\alpha}{8}\right\},$$

*where $\bar{\Psi}_0 := f(\bar{x}_0) - f(x^\star) + \lambda_0 \|\bar{g}_0 - \nabla f(\bar{x}_0)\|^2$, $\lambda_0 := \frac{b_0'}{4\eta_0(1-p_0)\mathcal{L}^2}$. Then, for any $\epsilon_f > 0$ Algorithm 2 returns a point $\bar{x}_K$ with $\mathbb{E}\left[f(\bar{x}_K) - f^\star\right] \leq \epsilon_f$ after $N := \sum_{k=0}^{K-1} T_k = \mathcal{O}\left(\epsilon_f^{-\frac{2-\alpha}{\alpha}}\right)$ iterations. The expected total computational cost (sample complexity) is*

$$cost := \sum_{k=0}^{K-1} T_k \left(p_k b_k + 2(1-p_k)b_k'\right) = \mathcal{O}\left(\epsilon_f^{-\frac{2}{\alpha}}\right).$$

*Proof.* Combining the result of Lemma 8 and Lemma 4 with $a = \frac{2}{\alpha}$, $x = \frac{\lambda G_t}{\Psi_t} \leq 1$, we obtain the following recursion

$$\Psi_{t+1} - \Psi_t \quad \leq \quad -\eta\mu\Psi_t^{\frac{2}{\alpha}} - \frac{p_k\lambda_k}{2}G_t\left(1 - \frac{4\eta\mu}{p_k\alpha}\Psi_t^{\frac{2-\alpha}{\alpha}}\right) + \frac{p_k\lambda_k}{2}\frac{\sigma^2}{b_k}, \tag{28}$$

where $\Psi_t := \delta_t + \lambda_k G_t$, $G_t := \mathbb{E}\left[\frac{1}{2}\|g_t - \nabla f(x_t)\|^2\right]$, $\delta_t := \mathbb{E}\left[f(x_t) - f(x^\star)\right]$, $\lambda_k := \frac{b_k'}{4\eta_k(1-p_k)\mathcal{L}^2}$.

Define the sequence $\{\bar{\Psi}_k\}_{k\geq 0}$ as $\bar{\Psi}_k := \mathbb{E}\left[f(\bar{x}_k) - f(x^\star) + \lambda_k\|\bar{g}_k - \nabla f(\bar{x}_k)\|^2\right]$, which corresponds to the outer loop of the Algorithm 2. For each $k = 0, \ldots, K-1$, the inner loop of Algorithm 2 starts with $x_0$ such that $\Psi_0 := \bar{\Psi}_k$. Let us prove by induction that within the outer loop $\bar{\Psi}_k \leq \frac{\bar{\Psi}_0}{2^k}$ for $k = 0, \ldots, K-1$ and, for each $k = 0, \ldots, K-1$, within the inner loop we have $\Psi_{t+1} \leq \Psi_t$ for $t = 0, \ldots T_k - 1$ (unless we reached the desired accuracy $\Psi_t \leq \frac{\bar{\Psi}_k}{2 \cdot 2^{\frac{2-\alpha}{\alpha}}}$ within the inner loop). The induction base for the outer loop and $k = 0$ is trivial. The induction base for the inner loop and $t = 0$ is verified by the assumption on the step-size and the choice of batch-sizes when $k = 0$. Fix $k = 0, \ldots, K-1$ and $t = 0, \ldots, T_k - 1$ and assume that we have $\Psi_t \leq \Psi_{t-1} \leq \Psi_0 = \bar{\Psi}_k$ and

$\bar{\Psi}_k \leq \frac{\bar{\Psi}_0}{2^k}$. Then it follows from (28) that

$$
\begin{aligned}
\Psi_{t+1} - \Psi_t \quad &\leq \quad -\eta\mu\Psi_t^{\frac{2}{\alpha}} - \frac{p_k\lambda_k}{2}G_t\left(1 - \frac{4\eta\mu}{p_k\alpha}\Psi_t^{\frac{2-\alpha}{\alpha}}\right) + \frac{p_k\lambda_k}{2}\frac{\sigma^2}{b_k} \\
&\leq \quad -\eta\mu\Psi_t^{\frac{2}{\alpha}} - \frac{p_k\lambda_k}{2}G_t\left(1 - \frac{4\eta\mu}{p_k\alpha}\bar{\Psi}_k^{\frac{2-\alpha}{\alpha}}\right) + \frac{p_k\lambda_k}{2}\frac{\sigma^2}{b_k} \\
&\leq \quad -\eta\mu\Psi_t^{\frac{2}{\alpha}} - \frac{p_k\lambda_k}{2}G_t\left(1 - \frac{4\eta\mu}{p_k\alpha}\left(\frac{\bar{\Psi}_0}{2^k}\right)^{\frac{2-\alpha}{\alpha}}\right) + \frac{p_k\lambda_k}{2}\frac{\sigma^2}{b_k} \\
&\overset{(i)}{\leq} \quad -\eta\mu\Psi_t^{\frac{2}{\alpha}} + \frac{p_k\lambda_k}{2}\frac{\sigma^2}{b_k} \\
&\overset{(ii)}{=} \quad -\eta\mu\Psi_t^{\frac{2}{\alpha}} + \frac{p_k}{2}\frac{\sigma^2}{b_k}\frac{b_k'}{4\eta(1-p_k)\mathcal{L}^2} \\
&\overset{(iii)}{=} \quad -\eta\mu\Psi_t^{\frac{2}{\alpha}} + \frac{\sigma^2}{b_k}\frac{1}{8\eta\mathcal{L}^2} \\
&\overset{(iv)}{=} \quad -\eta\mu\Psi_t^{\frac{2}{\alpha}} + \frac{\eta\mu}{2}\left(\frac{\bar{\Psi}_0}{2\cdot 2^{\frac{2-\alpha}{\alpha}}\cdot 2^k}\right)^{\frac{2}{\alpha}}.
\end{aligned}
$$

where $(i)$ follows by $p_k \geq \frac{1}{2b_k'} = \frac{4\eta\mu}{\alpha}\left(\frac{\bar{\Psi}_0}{2^k}\right)^{\frac{2-\alpha}{\alpha}}$ and the assumption on the step-size, $(ii)$ is due to $\lambda_k = \frac{b_k'}{4\eta(1-p_k)\mathcal{L}^2}$, $(iii)$ is due to $\frac{p_k b_k'}{1-p_k} = 1$, and $(iv)$ holds by the assumption on the batch-size $b_k$. The above recursion guaranties that after at most $T_k = \frac{2}{\eta\mu}\left(2\cdot 2^{\frac{2-\alpha}{\alpha}}\left(\frac{2^k U}{\bar{\Psi}_0} + 2\left(\frac{\eta\mu}{2}\right)^{\frac{\alpha}{2-\alpha}}\right)\right)^{\frac{2-\alpha}{\alpha}}$ inner loop iterations, we have $\Psi_{T_k} \leq \frac{\bar{\Psi}_0}{4} = \frac{\bar{\Psi}_k}{2\cdot 2^{\frac{2-\alpha}{\alpha}}} = \frac{\bar{\Psi}_0}{2\cdot 2^{\frac{2-\alpha}{\alpha}}\cdot 2^k}$. Indeed, if for $t = 0,\ldots,T_k - 1$, we have not reached $\Psi_t \leq \frac{\bar{\Psi}_0}{2\cdot 2^{\frac{2-\alpha}{\alpha}}\cdot 2^k}$, then $\Psi_{t+1} - \Psi_t \leq -\frac{\eta\mu}{2}\Psi_t^{\frac{2}{\alpha}} \leq 0$ and by Lemma 6 (with $c = \frac{2-\alpha}{\alpha}$, $b = \eta\mu/2$), we get $\Psi_{T_k} \leq \frac{\bar{\Psi}_0}{2\cdot 2^{\frac{2-\alpha}{\alpha}}} = \frac{\bar{\Psi}_k}{2\cdot 2^{\frac{2-\alpha}{\alpha}}}$. Now it remains to analyze the outer loop of Algorithm 2. By the definition of $\bar{\Psi}_k$ and the choice of batch-sizes $b_k'$ we have $\lambda_{k+1} \leq 2^{\frac{2-\alpha}{\alpha}}\lambda_k$ and $\bar{\Psi}_{k+1} \leq 2^{\frac{2-\alpha}{\alpha}}\Psi_{T_k} \leq \frac{\bar{\Psi}_k}{2} \leq \frac{\bar{\Psi}_0}{2^{k+1}}$. Thus, the induction step is complete.

In order to achieve $\bar{\Psi}_K \leq \epsilon_f$, we need $K = \log_2\left(\frac{\bar{\Psi}_0}{\epsilon_f}\right)$ outer loop iterations. The total number of iterations is

$$
\begin{aligned}
N \quad &= \quad \sum_{k=0}^{K-1} T_k \\
&= \quad \sum_{k=0}^{K-1}\frac{2}{\eta\mu}\left(2\cdot 2^{\frac{2-\alpha}{\alpha}}\left(\frac{2^k U}{\bar{\Psi}_0} + 2\left(\frac{\eta\mu}{2}\right)^{\frac{\alpha}{2-\alpha}}\right)\right)^{\frac{2-\alpha}{\alpha}} \\
&= \quad \frac{2}{\eta\mu}\left(2\cdot 2^{\frac{2-\alpha}{\alpha}}\right)^{\frac{2-\alpha}{\alpha}}\sum_{k=0}^{K-1}\left(\frac{2^k U}{\bar{\Psi}_0} + 2\left(\frac{\eta\mu}{2}\right)^{\frac{\alpha}{2-\alpha}}\right)^{\frac{2-\alpha}{\alpha}} \\
&= \quad \frac{2\cdot 2^{\frac{2(2-\alpha)}{\alpha^2}}}{\eta\mu}\left(\frac{U}{\bar{\Psi}_0} + 2\left(\frac{\eta\mu}{2}\right)^{\frac{\alpha}{2-\alpha}}\right)^{\frac{2-\alpha}{\alpha}}\sum_{k=0}^{K-1}\left(2^{\frac{2-\alpha}{\alpha}}\right)^k \\
&\leq \quad \frac{2\cdot 2^{\frac{2(2-\alpha)}{\alpha^2}}}{\eta\mu}\left(\frac{U}{\bar{\Psi}_0} + 2\left(\frac{\eta\mu}{2}\right)^{\frac{\alpha}{2-\alpha}}\right)^{\frac{2-\alpha}{\alpha}}\left(2^{\frac{2-\alpha}{\alpha}}\right)^K\left(2^{\frac{2-\alpha}{\alpha}} - 1\right)^{-1} \\
&\leq \quad \frac{2\cdot 2^{\frac{2(2-\alpha)}{\alpha^2}}}{\eta\mu}\left(\frac{U}{\bar{\Psi}_0} + 2\left(\frac{\eta\mu}{2}\right)^{\frac{\alpha}{2-\alpha}}\right)^{\frac{2-\alpha}{\alpha}}\left(2^{\frac{2-\alpha}{\alpha}} - 1\right)^{-1}\left(\frac{\bar{\Psi}_0}{\epsilon_f}\right)^{\frac{2-\alpha}{\alpha}}.
\end{aligned}
$$

The expected computational cost per iteration is

$$
\begin{aligned}
p_k b_k + 2(1-p_k) b'_k \ &\le\ \frac{b_k}{1+b'_k} + 2b'_k \\[2mm]
&\le\ \frac{b_k}{b'_k} + 2b'_k \\[2mm]
&\le\ \frac{\left(\frac{2\cdot 2^{\frac{2-\alpha}{\alpha}}\cdot 2^k}{\overline{\Psi}_0}\right)^{\frac{2}{\alpha}}\frac{\sigma^2}{4\mu\eta^2\mathcal{L}^2}}{\frac{\alpha}{8\eta\mu}\left(\frac{2^k}{\overline{\Psi}_0}\right)^{\frac{2-\alpha}{\alpha}}} + 2\frac{\alpha}{8\eta\mu}\left(\frac{2^k}{\overline{\Psi}_0}\right)^{\frac{2-\alpha}{\alpha}} \\[2mm]
&\le\ \left(2\cdot 2^{\frac{2-\alpha}{\alpha}}\right)^{\frac{2}{\alpha}}\frac{2\sigma^2}{\eta\mathcal{L}^2}\frac{2^k}{\overline{\Psi}_0} + \frac{\alpha}{4\eta\mu}\left(\frac{2^k}{\overline{\Psi}_0}\right)^{\frac{2-\alpha}{\alpha}} \\[2mm]
&\le\ \frac{2\sigma^2\cdot 2^{4/\alpha^2}}{4\eta\mathcal{L}^2}\frac{2^k}{\overline{\Psi}_0} + \frac{\alpha}{4\eta\mu}\left(\frac{2^k}{\overline{\Psi}_0}\right)^{\frac{2-\alpha}{\alpha}} \\[2mm]
&\le\ \left(\frac{\sigma^2\cdot 2^{4/\alpha^2}}{4\eta\mathcal{L}^2\overline{\Psi}_0} + \frac{\alpha}{4\eta\mu\overline{\Psi}_0^{\frac{2-\alpha}{\alpha}}}\right)2^k.
\end{aligned}
$$

Denote $A := \left(\frac{\sigma^2\cdot 2^{4/\alpha^2}}{4\eta\mathcal{L}^2\overline{\Psi}_0} + \frac{\alpha}{4\eta\mu\overline{\Psi}_0^{\frac{2-\alpha}{\alpha}}}\right)$, then the total cost is

$$
\begin{aligned}
\text{cost}\ &=\ \sum_{k=0}^{K-1} T_k\left(p_k b_k + 2(1-p_k)b'_k\right) \\[2mm]
&=\ A\sum_{k=0}^{K-1} T_k\cdot 2^k \\[2mm]
&=\ A\sum_{k=0}^{K-1}\frac{2}{\eta\mu}\left(2\cdot 2^{\frac{2-\alpha}{\alpha}}\left(\frac{2^k U}{\overline{\Psi}_0} + 2\left(\frac{\eta\mu}{2}\right)^{\frac{\alpha}{2-\alpha}}\right)\right)^{\frac{2-\alpha}{\alpha}}2^k \\[2mm]
&=\ 2\cdot 2^{\frac{2(2-\alpha)}{\alpha^2}}A\left(\frac{2}{\eta\mu}\left(\frac{U}{\overline{\Psi}_0}\right)^{\frac{2-\alpha}{\alpha}}\sum_{k=0}^{K-1}\left(2^k\right)^{\frac{2-\alpha}{\alpha}}2^k + 2^{\frac{2-\alpha}{\alpha}}\sum_{k=0}^{K-1}2^k\right) \\[2mm]
&=\ 2\cdot 2^{\frac{2(2-\alpha)}{\alpha^2}}A\left(\frac{2}{\eta\mu}\left(\frac{U}{\overline{\Psi}_0}\right)^{\frac{2-\alpha}{\alpha}}\sum_{k=0}^{K-1}\left(2^k\right)^{\frac{2}{\alpha}} + 2^{\frac{2-\alpha}{\alpha}}\sum_{k=0}^{K-1}2^k\right) \\[2mm]
&=\ 2\cdot 2^{\frac{2(2-\alpha)}{\alpha^2}}A\left(\frac{2}{\eta\mu}\left(\frac{U}{\overline{\Psi}_0}\right)^{\frac{2-\alpha}{\alpha}}\left(2^K\right)^{\frac{2}{\alpha}}\left(2^{2/\alpha}-1\right)^{-1} + 2^{\frac{2-\alpha}{\alpha}}2^K\right),
\end{aligned}
$$

which further simplifies by using the value of $A$ and the step-size

$$
\begin{aligned}
\text{cost} \quad &= \quad \mathcal{O}\left( \frac{A}{\eta\mu} \left( \frac{1}{\bar{\Psi}_0} \right)^{\frac{2-\alpha}{\alpha}} \left( 2^K \right)^{\frac{2}{\alpha}} \right) \\
&= \quad \mathcal{O}\left( \frac{A}{\eta\mu} \left( \frac{1}{\bar{\Psi}_0} \right)^{\frac{2-\alpha}{\alpha}} \left( \frac{\bar{\Psi}_0}{\epsilon_f} \right)^{\frac{2}{\alpha}} \right) \\
&= \quad \mathcal{O}\left( \frac{A\bar{\Psi}_0}{\eta\mu} \left( \frac{1}{\epsilon_f} \right)^{\frac{2}{\alpha}} \right) \\
&= \quad \mathcal{O}\left( \left( \frac{\sigma^2}{\mu} + \frac{\bar{\Psi}_0^{\frac{2(\alpha-1)}{\alpha}}}{\eta^2\mu^2} \right) \left( \frac{1}{\epsilon_f} \right)^{\frac{2}{\alpha}} \right) \\
&= \quad \mathcal{O}\left( \left( \frac{\sigma^2}{\mu} + \kappa^2 \bar{\Psi}_0^{\frac{2(\alpha-1)}{\alpha}} \right) \left( \frac{1}{\epsilon_f} \right)^{\frac{2}{\alpha}} \right) \\
&= \quad \mathcal{O}\left( \epsilon_f^{-2/\alpha} \right).
\end{aligned}
$$

$\square$

## C.2 Proof of Theorem 3

Now we state and prove a detailed version of Theorem 3.

**Theorem 5.** *Let $f(\cdot)$ have the form (2) and satisfy Assumptions 1, 3 (with $\alpha \in [1,2)$) and 6, let the sequences in Algorithm 2 be chosen as $p_k = \frac{1}{n+1}$, $b'_k = 1$, $b_k = n$,*

$$
T_k = \frac{1}{\eta_k\mu} \left( \frac{U 2^{k+1}}{\bar{\Psi}_0} + 2 \left( \eta_k\mu \right)^{\frac{\alpha}{2-\alpha}} \right)^{\frac{2-\alpha}{\alpha}},
$$

$$
\eta_k = \min\left\{ \frac{1}{2\sqrt{n}\mathcal{L}}, \frac{\alpha}{4\mu(n+1)} \left( \frac{2^k}{\bar{\Psi}_0} \right)^{\frac{2-\alpha}{\alpha}} \right\},
$$

*where $\bar{\Psi}_0 := f(\bar{x}_0) - f(x^\star) + \lambda_0 \|\bar{g}_0 - \nabla f(\bar{x}_0)\|^2$, $\lambda_0 := \frac{b'}{4\eta_0(1-p)\mathcal{L}^2}$ Then, for any $\epsilon_f > 0$, Algorithm 2 returns a point $\bar{x}_K$ with $\mathbb{E}\left[ f(\bar{x}_K) - f^\star \right] \le \epsilon_f$ after*

$$
N \quad := \quad \sum_{k=0}^{K-1} T_k = \widetilde{\mathcal{O}}\left( n + \sqrt{n}\kappa\epsilon_f^{-\frac{2-\alpha}{\alpha}} \right)
$$

*iterations. The expected total computational cost (sample complexity) is*

$$
\text{cost} := \sum_{k=0}^{K-1} T_k \left( p_k b_k + 2(1-p_k)b'_k \right) = \widetilde{\mathcal{O}}\left( n + \sqrt{n}\kappa\epsilon_f^{-\frac{2-\alpha}{\alpha}} \right).
$$

*Proof.* Combining the result of Lemma 8 and Lemma 4 with $a = \frac{2}{\alpha}$, $x = \frac{\lambda G_t}{\Psi_t} \le 1$ and noticing that $\sigma^2 = 0$, we obtain the following recursion

$$
\Psi_{t+1} - \Psi_t \quad \le \quad -\eta\mu\Psi_t^{\frac{2}{\alpha}} - \frac{p\lambda}{2} G_t \left( 1 - \frac{4\eta\mu(n+1)}{\alpha} \Psi_t^{\frac{2-\alpha}{\alpha}} \right), \quad\quad\quad (29)
$$

where $\Psi_t := \delta_t + \lambda_k G_t$, $G_t := \mathbb{E}\left[ \frac{1}{2} \|g_t - \nabla f(x_t)\|^2 \right]$, $\delta_t := \mathbb{E}\left[ f(x_t) - f(x^\star) \right]$, $\lambda_k := \frac{b'}{4\eta_k(1-p)\mathcal{L}^2}$.

Define the sequence $\left\{ \bar{\Psi}_k \right\}_{k\ge0}$ as $\bar{\Psi}_k := \mathbb{E}\left[ f(\bar{x}_k) - f(x^\star) + \lambda_k \|\bar{g}_k - \nabla f(\bar{x}_k)\|^2 \right]$ and $\lambda_k := \frac{b'}{4\eta_k(1-p)\mathcal{L}^2}$, which corresponds to the outer loop of the algorithm. For each $k = 0, \ldots, K-1$,

the inner loop of Algorithm 2 starts with $x_0$ such that $\Psi_0 := \bar{\Psi}_k$. Let us prove by induction that the sequence $\{\bar{\Psi}_k\}_{k\geq 0}$ satisfies $\bar{\Psi}_k \leq \frac{\bar{\Psi}_0}{2^k}$ for all $k = 0, \ldots, K-1$. The induction base for $k = 0$ is trivial. Let us prove the induction step for $k+1$. The evolution of the inner loop is characterized by (34) and given the assumption on the step-size, we have $\Psi_{t+1} - \Psi_t \leq -\eta\mu\Psi_t^{\frac{2}{\alpha}}$ for all $t = 0, \ldots, T_k - 1$. Therefore, by Lemma 6 (with $c = \frac{2-\alpha}{\alpha}$, $b = \eta\mu$) we have

$$
\begin{aligned}
\Psi_{T_k} &\leq \frac{U + (\eta_k\mu)^{\frac{\alpha}{2-\alpha}}\bar{\Psi}_k}{(\eta_k\mu T_k)^{\frac{\alpha}{2-\alpha}}} = \frac{U + (\eta_k\mu)^{\frac{\alpha}{2-\alpha}}\bar{\Psi}_k}{\frac{U\cdot 2^{k+1}}{\bar{\Psi}_0} + 2\left(\eta_k\mu\right)^{\frac{\alpha}{2-\alpha}}} \\
&= \frac{U + (\eta_k\mu)^{\frac{\alpha}{2-\alpha}}\bar{\Psi}_k}{U + (\eta_k\mu)^{\frac{\alpha}{2-\alpha}}\frac{\bar{\Psi}_0}{2^k}} \cdot \frac{\bar{\Psi}_0}{2^{k+1}} \overset{(i)}{\leq} \frac{\bar{\Psi}_0}{2^{k+1}},
\end{aligned}
$$

where in $(i)$, we used $\bar{\Psi}_k \leq \frac{\bar{\Psi}_0}{2^k}$. Furthermore, since $\eta_{k+1} \geq \eta_k$, then $\lambda_{k+1} \leq \lambda_k$ and $\bar{\Psi}_{k+1} \leq \Psi_{T_k} \leq \frac{\bar{\Psi}_0}{2^{k+1}}$, and the induction step is complete.

In order to achieve $\bar{\Psi}_K \leq \epsilon_f$, we need $K = \log_2\left(\frac{\bar{\Psi}_0}{\epsilon_f}\right)$ outer loop iterations. The total number of iterations is

$$
\begin{aligned}
N &= \sum_{k=0}^{K-1} T_k \\
&\overset{(i)}{\leq} \sum_{k=0}^{K-1} \max\left\{\frac{4(n+1)}{\alpha}\left(\frac{\bar{\Psi}_0}{2^k}\right)^{\frac{(2-\alpha)}{\alpha}}, 2\sqrt{n}\kappa\right\}\left(\frac{U2^{k+1}}{\bar{\Psi}_0} + \frac{\mu}{\sqrt{n}\mathcal{L}}\right)^{\frac{2-\alpha}{\alpha}} \\
&\leq \sum_{k=0}^{K-1} \max\left\{\frac{4(n+1)}{\alpha}\left(2\left(U + \frac{\bar{\Psi}_0}{\sqrt{n}\kappa}\right)\right)^{\frac{2-\alpha}{\alpha}}, 2\sqrt{n}\kappa\left(\frac{2U}{\bar{\Psi}_0} + \frac{1}{\sqrt{n}\kappa}\right)^{\frac{2-\alpha}{\alpha}}\left(2^{\frac{2-\alpha}{\alpha}}\right)^k\right\} \\
&\leq \max\left\{\frac{4(n+1)}{\alpha}\left(2\left(U + \frac{\bar{\Psi}_0}{\sqrt{n}\kappa}\right)\right)^{\frac{2-\alpha}{\alpha}}K, 2\sqrt{n}\kappa\left(\frac{2U}{\bar{\Psi}_0} + \frac{1}{\sqrt{n}\kappa}\right)^{\frac{2-\alpha}{\alpha}}\left(2^{\frac{2-\alpha}{\alpha}}\right)^K\left(2^{\frac{2-\alpha}{\alpha}} - 1\right)^{-1}\right\} \\
&\leq \max\left\{\frac{4(n+1)}{\alpha}\left(2\left(U + \frac{\bar{\Psi}_0}{\sqrt{n}\kappa}\right)\right)^{\frac{2-\alpha}{\alpha}}\log_2\left(\frac{\bar{\Psi}_0}{\epsilon_f}\right), \frac{2\sqrt{n}}{2^{\frac{2-\alpha}{\alpha}} - 1}\kappa\left(\frac{2U}{\bar{\Psi}_0} + \frac{1}{\sqrt{n}\kappa}\right)^{\frac{2-\alpha}{\alpha}}\left(\frac{\bar{\Psi}_0}{\epsilon_f}\right)^{\frac{2-\alpha}{\alpha}}\right\} \\
&= \widetilde{\mathcal{O}}\left(n + \sqrt{n}\kappa\epsilon_f^{-\frac{2-\alpha}{\alpha}}\right),
\end{aligned}
$$

where in $(i)$ we used the assumption on the step-sizes. The expected computational cost per iteration is $p_k b_k + 2(1-p_k)b'_k \leq 3$ and thus the total cost is $\widetilde{\mathcal{O}}\left(n + \sqrt{n}\kappa\epsilon_f^{-\frac{2-\alpha}{\alpha}}\right)$. $\qquad\square$

### C.3 Technical lemmas

**Lemma 4.** *Let $x \leq 1$ and $a \geq 1$, then $(1-x)^a \geq 1 - ax$.*

*Proof.* The results follows directly by applying the definition of convexity. $\qquad\square$

The following lemma is standard, we provide its proof for completeness.

**Lemma 5.** *Suppose that function $f(\cdot)$ is L-smooth and let $x_{t+1} := x_t - \eta g_t$, where $g_t \in \mathbb{R}^d$ is any vector, and $\eta > 0$ any scalar. Then we have*

$$
f(x_{t+1}) \leq f(x_t) - \frac{\eta}{2}\|\nabla f(x_t)\|^2 - \left(\frac{1}{2\eta} - \frac{L}{2}\right)\|x_{t+1} - x_t\|^2 + \frac{\eta}{2}\|g_t - \nabla f(x_t)\|^2. \tag{30}
$$

*Proof.* Define $\bar{x}_{t+1} := x_t - \eta \nabla f(x_t)$, then using Assumption 1 after some rearrangements we obtain

$$
\begin{aligned}
f(x_{t+1}) \quad \leq \quad & f(x_t) + \langle \nabla f(x_t), x_{t+1} - x_t \rangle + \frac{L}{2} \|x_{t+1} - x_t\|^2 \\
= \quad & f(x_t) + \langle \nabla f(x_t) - g_t, x_{t+1} - x_t \rangle + \langle g_t, x_{t+1} - x_t \rangle + \frac{L}{2} \|x_{t+1} - x_t\|^2 \\
= \quad & f(x_t) + \langle \nabla f(x_t) - g_t, -\eta g_t \rangle - \left( \frac{1}{\eta} - \frac{L}{2} \right) \|x_{t+1} - x_t\|^2 \\
= \quad & f(x_t) + \eta \|\nabla f(x_t) - g_t\|^2 - \eta \langle \nabla f(x_t) - g_t, \nabla f(x_t) \rangle - \left( \frac{1}{\eta} - \frac{L}{2} \right) \|x_{t+1} - x_t\|^2 \\
= \quad & f(x_t) + \eta \|\nabla f(x_t) - g_t\|^2 - \frac{1}{\eta} \langle x_{t+1} - \bar{x}_{t+1}, x_t - \bar{x}_{t+1} \rangle - \left( \frac{1}{\eta} - \frac{L}{2} \right) \|x_{t+1} - x_t\|^2 \\
= \quad & f(x_t) + \eta \|\nabla f(x_t) - g_t\|^2 - \left( \frac{1}{\eta} - \frac{L}{2} \right) \|x_{t+1} - x_t\|^2 \\
& - \frac{1}{2\eta} \left( \|x_{t+1} - \bar{x}_{t+1}\|^2 + \|x_t - \bar{x}_{t+1}\|^2 - \|x_{t+1} - x_t\|^2 \right) \\
= \quad & f(x_t) + \eta \|\nabla f(x_t) - g_t\|^2 - \left( \frac{1}{\eta} - \frac{L}{2} \right) \|x_{t+1} - x_t\|^2 \\
& - \frac{1}{2\eta} \left( \eta^2 \|\nabla f(x_t) - g_t\|^2 + \eta^2 \|\nabla f(x_t)\|^2 - \|x_{t+1} - x_t\|^2 \right) \\
= \quad & f(x_t) - \frac{\eta}{2} \|\nabla f(x_t)\|^2 - \left( \frac{1}{2\eta} - \frac{L}{2} \right) \|x_{t+1} - x_t\|^2 + \frac{\eta}{2} \|g_t - \nabla f(x_t)\|^2.
\end{aligned}
$$

$\square$

**Lemma 6.** *Let $\{r_k\}_{k \geq 0}$ be a non-negative sequence, which satisfies*

$$
r_{k+1} \leq r_k(1 - b r_k^c), \quad \text{for all } k
$$

*and $c > 0$. Then*

$$
r_k \leq \frac{U + b^{1/c} r_0}{(b(k+1))^{1/c}},
$$

*where $U := 2^{1/c} \cdot c^{-\frac{2}{c} - 1} + c^{-1/c}$.*

*Proof.* Define $u_k := \varphi(k) r_k$, $\varphi(k) := (b(k+1))^{1/c}$. Then using $\varphi(k+1) - \varphi(k) \leq \frac{1}{c} \frac{\varphi(k+1)}{k+2}$ and $1 \leq \varphi(k+1)(\varphi(k))^{-1} \leq 2^{1/c}$, we obtain

$$
\begin{aligned}
u_{k+1} - u_k \quad = \quad & \varphi(k+1) r_{k+1} - \varphi(k) r_k \\
\leq \quad & (\varphi(k+1) - \varphi(k)) r_k - b \varphi(k+1) r_k^{1+c} \\
= \quad & (\varphi(k+1) - \varphi(k)) (\varphi(k))^{-1} u_k - b \varphi(k+1) (\varphi(k))^{-1-c} u_k^{1+c} \\
= \quad & (\varphi(k+1) - \varphi(k)) (\varphi(k))^{-1} u_k \left( 1 - \frac{\varphi(k+1) u_k^c}{(k+1)(\varphi(k+1) - \varphi(k))} \right) \\
\leq \quad & (\varphi(k+1) - \varphi(k)) (\varphi(k))^{-1} u_k (1 - c u_k^c).
\end{aligned}
$$

It follows from the above recursion that the sequence $\{u_k\}_{k \geq 0}$ is bounded for all $k$. Indeed, define $F(k, u) := (\varphi(k+1) - \varphi(k)) (\varphi(k))^{-1} u (1 - c u^c)$. Notice that for all $k \geq 0$ and $u > c^{-1/c}$ we have $F(k, u) < 0$ and for all $k, u \geq 0$ we have $F(k, u) \leq 2^{1/c} \cdot c^{-\frac{2}{c} - 1}$. Now it is straightforward to see that $u_k \leq u_0 + 2^{1/c} \cdot c^{-\frac{2}{c} - 1} + c^{-1/c}$. It only remains to return to $r_k$ sequence to obtain the desired result. $\square$

**Lemma 7** (Lemma 4 of [36]). *Let Assumptions 5 and 6 hold, and let for $\chi \sim$ Bernoulli$(p)$ and $g_t \in \mathbb{R}^d$, we construct $g_{t+1}$ via*

$$g_{t+1} = \begin{cases} \frac{1}{b}\sum_{i=1}^{b} \nabla f_{\xi_{t+1}^i}(x_{t+1}) & \text{if} \quad \chi = 1, \\ g_t + \frac{1}{b'}\sum_{i=1}^{b'}\left(\nabla f_{\xi_{t+1}^i}(x_{t+1}) - \nabla f_{\xi_{t+1}^i}(x_t)\right) & \text{if} \quad \chi = 0. \end{cases} \tag{31}$$

*Then*

$$G_{t+1} - G_t \leq -pG_t + \frac{(1-p)\mathcal{L}^2}{b'}R_t + \frac{p\sigma^2}{2b}, \tag{32}$$

*where $G_t := \mathbb{E}\left[\frac{1}{2}\|g_t - \nabla f(x_t)\|^2\right]$, $R_n := \mathbb{E}\left[\frac{1}{2}\|x_{t+1} - x_t\|^2\right]$.*

*Proof.*

$$
\begin{aligned}
G_{t+1} &= \mathbb{E}\left[\frac{1}{2}\|g_{t+1} - \nabla f(x_{t+1})\|^2\right] \\
&= p\mathbb{E}\left[\frac{1}{2}\left\|\frac{1}{b}\sum_{i=1}^{b}\nabla f_{\xi_{t+1}^i}(x_{t+1}) - \nabla f(x_{t+1})\right\|^2\right] \\
&\quad + (1-p)\,\mathbb{E}\left[\frac{1}{2}\left\|g_t + \frac{1}{b'}\sum_{i=1}^{b'}\left(\nabla f_{\xi_{t+1}^i}(x_{t+1}) - \nabla f_{\xi_{t+1}^i}(x_t)\right) - \nabla f(x_{t+1})\right\|^2\right] \\
&\leq \frac{p\sigma^2}{2b} + (1-p)\,\mathbb{E}\left[\frac{1}{2}\left\|g_t + \frac{1}{b'}\sum_{i=1}^{b'}\left(\nabla f_{\xi_{t+1}^i}(x_{t+1}) - \nabla f_{\xi_{t+1}^i}(x_t)\right) - \nabla f(x_{t+1})\right\|^2\right] \\
&= \frac{p\sigma^2}{2b} + (1-p)\,\mathbb{E}\left[\frac{1}{2}\left\|g_t - \nabla f(x_t) + \widetilde{\Delta}(x_{t+1}, x_t) - \Delta(x_{t+1}, x_t)\right\|^2\right] \\
&= \frac{p\sigma^2}{2b} + (1-p)\,\mathbb{E}\left[\frac{1}{2}\|g_t - \nabla f(x_t)\|^2\right] + (1-p)\,\mathbb{E}\left[\frac{1}{2}\left\|\widetilde{\Delta}(x_{t+1}, x_t) - \Delta(x_{t+1}, x_t)\right\|^2\right] \\
&\leq (1-p)\,\mathbb{E}\left[\frac{1}{2}\|g_t - \nabla f(x_t)\|^2\right] + \frac{(1-p)\mathcal{L}^2}{b'}\mathbb{E}\left[\frac{1}{2}\|x_{t+1} - x_t\|^2\right] + \frac{p\sigma^2}{2b} \\
&= (1-p)\,G_t + \frac{(1-p)\mathcal{L}^2}{b'}R_t + \frac{p\sigma^2}{2b},
\end{aligned}
$$

where the first inequality holds by Assumption 5 and the second inequality is due to Assumption 6 with $\widetilde{\Delta}(x,y) := \frac{1}{b'}\sum_{i=1}^{b'}\left(\nabla f_{\xi_{t+1}^i}(x) - \nabla f_{\xi_{t+1}^i}(y)\right)$, $\Delta(x,y) := \nabla f(x) - \nabla f(y)$, $x = x_{t+1}$, $y = x_t$. $\square$

**Lemma 8.** *Let $f(\cdot)$ satisfy Assumptions 1, 3, 5 and 6. Assume that the step-size in Algorithm 3 satisfies*

$$\eta \leq \min\left\{\frac{1}{2L}, \sqrt{\frac{pb'}{1-p}}\frac{1}{2\mathcal{L}}\right\}. \tag{33}$$

*Define $\Psi_t := \mathbb{E}\left[f(x_t) - f(x^\star) + \lambda\|g_t - \nabla f(x_t)\|^2\right]$, $\lambda := \frac{b'}{4\eta(1-p)\mathcal{L}^2}$. Then Algorithm 3 generates a sequence of points $\{x_t\}_{t\geq 0}$ such that*

$$\Psi_{t+1} - \Psi_t \leq -\eta\mu\left(\Psi_t - \lambda G_t\right)^{\frac{2}{\alpha}} - \frac{p\lambda}{2}G_t + \frac{p\lambda}{2}\frac{\sigma^2}{b}. \tag{34}$$

*Proof.* Using the notation $G_t := \mathbb{E}\left[\frac{1}{2}\|g_t - \nabla f(x_t)\|^2\right]$, $R_t := \mathbb{E}\left[\frac{1}{2}\|x_{t+1} - x_t\|^2\right]$, $\delta_t := \mathbb{E}\left[f(x_t) - f(x^\star)\right]$ and assumption on the step-size $\eta \leq \frac{1}{2L}$, it follows by Lemma 5 that

$$\delta_{t+1} - \delta_t \leq -\frac{\eta}{2}\mathbb{E}\left[\|\nabla f(x_t)\|^2\right] - \frac{1}{2\eta}R_t + \eta G_t.$$

Using Assumption 3, Jensen's inequality for $x \mapsto x^{\frac{2}{\alpha}}$, we get

$$\delta_{t+1} - \delta_t \leq -\eta\mu\delta_t^{\frac{2}{\alpha}} + \eta G_t - \frac{1}{2\eta}R_t.$$

For $p < 1$, it follows from Lemma 7 that

$$-R_t \leq -\frac{b'}{(1-p)\mathcal{L}^2}\left(G_{t+1} - G_t\right) - \frac{pb'}{(1-p)\mathcal{L}^2}G_t + \frac{b'}{(1-p)\mathcal{L}^2}\frac{p\sigma^2}{2b}.$$

Thus, combining the above two inequalities, we get

$$\delta_{t+1} - \delta_t + \frac{1}{2\eta}\frac{b'}{(1-p)\mathcal{L}^2}\left(G_{t+1} - G_t\right) \leq -\eta\mu\delta_t^{\frac{2}{\alpha}} - \left(\frac{1}{2\eta}\frac{pb'}{(1-p)\mathcal{L}^2} - \eta\right)G_t + \frac{1}{2\eta}\frac{b'}{(1-p)\mathcal{L}^2}\frac{p\sigma^2}{2b}.$$

Let $\Psi_t := \delta_t + \lambda G_t$, $\lambda := \frac{b'}{2\eta(1-p)\mathcal{L}^2}$. Using the assumption on the step-size, $\eta \leq \sqrt{\frac{pb'}{4(1-p)\mathcal{L}^2}}$, we get

$$
\begin{aligned}
\Psi_{t+1} - \Psi_t &= \delta_{t+1} - \delta_t + \lambda\left(G_{t+1} - G_t\right) \\
&= \delta_{t+1} - \delta_t + \frac{b'}{2\eta(1-p)\mathcal{L}^2}\left(G_{t+1} - G_t\right) \\
&\leq -\eta\mu\delta_t^{\frac{2}{\alpha}} - \frac{pb'}{4\eta(1-p)\mathcal{L}^2}G_t + \frac{1}{2\eta}\frac{b'}{(1-p)\mathcal{L}^2}\frac{p\sigma^2}{2b} \\
&= -\eta\mu\delta_t^{\frac{2}{\alpha}} - \frac{p\lambda}{2}G_t + \frac{p\lambda}{2}\frac{\sigma^2}{b} \\
&= -\eta\mu\left(\Psi_t - \lambda G_t\right)^{\frac{2}{\alpha}} - \frac{p\lambda}{2}G_t + \frac{p\lambda}{2}\frac{\sigma^2}{b}.
\end{aligned}
$$

$\square$

# D   Convergence in the Iterates

In this Section, we assume that $\alpha$-PŁ condition holds with $\alpha \in (1, 2]$. We provide convergence guaranties in the *iterates* to the set of optimal points $X^\star$, which we assume to be non-empty. The sample complexity results are summarized in Table 2. The results in Table 2 are obtained by translating the sample complexity results reported in Table 1 to convergence in the iterates via Proposition 4. Note that in the special case $\alpha = 2$, our rates in both Tables 1 and 2 recover the optimal rates for online case [26, 23, 30] and the best known results for finite sum case [49, 36]. [14]

**Proposition 4.** *Let Assumption 3 hold with $\alpha \in (1, 2]$ and the set of optimal points $X^\star := \arg\min_x f(x)$ is not empty. Then*

$$dist\left(x, X^\star\right) \leq \frac{\alpha}{\alpha - 1}\frac{1}{\sqrt{2\mu}}\left(f(x) - f^\star\right)^{\frac{\alpha-1}{\alpha}} \quad \text{for all } x \in \mathbb{R}^d, \tag{35}$$

*where $dist\left(x, X^\star\right) := \min_{y \in X^\star}\|y - x\|$.*

The above result can be obtained by following the argument similar to the proof of Theorem 2 in [26] (where it is shown for a particular case $\alpha = 2$). The only difference is that one should take a disingularizing function as $g(x) = \left(f(x) - f^\star\right)^{\frac{\alpha-1}{\alpha}}$, where $f^\star = \min_x f(x)$. This result immediately implies convergence in the iterates via

$$
\begin{aligned}
\mathbb{E}\left[\min_{y \in X^\star}\|x - y\|\right] &= \mathbb{E}\left[dist\left(x, X^\star\right)\right] \\
&\overset{(35)}{\leq} \frac{\alpha}{\alpha - 1}\frac{1}{\sqrt{2\mu}}\mathbb{E}\left[\left(f(x) - f^\star\right)^{\frac{\alpha-1}{\alpha}}\right] \\
&\leq \frac{\alpha}{\alpha - 1}\frac{1}{\sqrt{2\mu}}\left(\mathbb{E}\left[f(x) - f^\star\right]\right)^{\frac{\alpha-1}{\alpha}}, \tag{36}
\end{aligned}
$$

where the last inequality holds by Jensen's inequality for a concave function $t \mapsto t^{\frac{\alpha-1}{\alpha}}$.

---

[14]While our analysis for variance reduction formally holds for $\alpha < 2$ only, the special case $\alpha = 2$ can be easily recovered via standard techniques, e.g., [49, 36].

Table 2: Summary of sample complexity results for $\alpha$-PŁ functions (Assumption 3) with $\alpha \in (1, 2]$ under average $\mathcal{L}$-smoothness (Assumptions 6) and bounded variance (Assumptions 5). Quantities: $\alpha$ = PL power; $\mu$ = PL constant; $\kappa = \mathcal{L}/\mu$; $\sigma^2$ = variance. The entries of the table show the expected number of stochastic gradient calls to achieve $\mathbb{E}\left[dist\left(x, X^\star\right)\right] \le \epsilon_x$, where $X^\star \neq \emptyset$ is the set of optimal points of $f(\cdot)$.

| Method | Finite sum case | Online case |
|--------|-----------------|-------------|
| GD | $\mathcal{O}\left(n\kappa\mu^{\frac{\alpha-2}{2(\alpha-1)}}\left(\frac{1}{\epsilon_x}\right)^{\frac{2-\alpha}{\alpha-1}}\right)$ | N/A |
| SGD | $\mathcal{O}\left(\kappa\sigma^2\mu^{\frac{\alpha+2}{2(1-\alpha)}}\left(\frac{1}{\epsilon_x}\right)^{\frac{4-\alpha}{\alpha-1}}\right)$ | $\mathcal{O}\left(\kappa\sigma^2\mu^{\frac{\alpha+2}{2(1-\alpha)}}\left(\frac{1}{\epsilon_x}\right)^{\frac{4-\alpha}{\alpha-1}}\right)$ |
| PAGER | $\widetilde{\mathcal{O}}\left(n + \sqrt{n}\kappa\mu^{\frac{\alpha-2}{2(\alpha-1)}}\left(\frac{1}{\epsilon_x}\right)^{\frac{2-\alpha}{\alpha-1}}\right)$ (new) | $\mathcal{O}\left(\left(\frac{\sigma^2}{\mu}+\kappa^2\right)\mu^{\frac{1}{1-\alpha}}\left(\frac{1}{\epsilon_f}\right)^{\frac{2}{\alpha-1}}\right)$ (new) |

## E   Simulations

In this section, we perform numerical tests to evaluate the performance of the discussed methods. Our experiments are based on the RL setup described in Example 5 since we believe that it is one of the most interesting applications of our theoretical results. The goal of our experiments is twofold. First, we want to make sure that variance reduction technique is useful in maximizing a cumulative reward for policy optimization tasks. Second, it is interesting to find out if the restarting procedure in PAGER is helpful in practice.

**Algorithmic adjustments.** In order to make Algorithms 1 and 2 applicable to the setup of Example 5, one needs to make some standard adjustments. First, we should specify the way the stochastic gradient is computed. In our experiments, we use the standard GPOMDP estimator [5], which is given by

$$g_k(\theta, \tau) := \frac{1}{b_k}\sum_{i=1}^{b_k}\sum_{h=0}^{H-1}\gamma^h r(s_h^i, a_h^i)Z_{\theta,h},$$

where $Z_{\theta,h} := \sum_{z=0}^{h}\nabla_\theta \log \pi_\theta(a_z^i|s_z^i)$, $\tau := \left\{(s_h^i, a_h^i)\right\}_{h=0}^{H-1}$ is generated according to the trajectory distribution $p(\tau|\pi_\theta)$, $\pi_\theta$ is the parametric policy and $H$ is the horizon length of an episode. Second, the data distribution changes over iterations (distribution shift), and one needs to use an importance weighting technique in order to apply variance reduction methods [47]. Importance weighting is implemented as

$$g'_{k,\omega_{\theta_2}}(\theta_1, \tau) := \frac{1}{b'_k}\sum_{i=1}^{b'_k}\omega(\tau_i|\theta_2, \theta_1)\sum_{h=0}^{H-1}\gamma^h r(s_h^i, a_h^i)Z_{\theta,h} \qquad \omega(\tau_i|\theta_2, \theta_1) := \Pi_{j=0}^{H-1}\frac{\pi_{\theta_1}(a_j^i|s_j^i)}{\pi_{\theta_2}(a_j^i|s_j^i)}.$$

Given the above notation PAGE gradient estimator can be computed as

$$g_{t+1} = \begin{cases} g_k(\theta_{t+1}, \tau_{t+1}), & \text{w.p.} \quad p, \\ g_t + g'_k(\theta_{t+1}, \tau_{t+1}) - g'_{k,\omega_{\theta_{t+1}}}(\theta_t, \tau_t), & \text{w.p. } 1-p. \end{cases}$$

**Experimental setup.** We test the discussed methods on benchmark RL environments CartPole and Acrobot that are available on OpenAI gym [12]. Both environments have discrete action space and continuous state space. We use a neural network with two hidden layers of width 32 each and Tanh activation function. We set parameters by default as $H = 200$, $\gamma = 0.9999$ and initialize all runs with the same randomly generated policy. For SGD, we use $T = 1$, $b = 50$. For PAGE we use $b = 50$, $b' = 5$, $p = 0.1$. For PAGER, we set initial batch-sizes as $b'_0 = 15$, $b_0 = 5$, $p_0 = 1$ $T_0 = 50$ and change the values from one stage to another based the formulas given by Theorem 2 (with $\alpha = 1$). We tune step-sizes from the set $\left\{10^{-5}, 2\cdot10^{-5}, \dots 2^6\cdot10^{-5}\right\}$ and select the one that gives the best performance based on the average reward in the last 10 iterations. The convergence curves Figure 4 are calculated as the mean over multiple runs with fixed parameters, the shaded regions represent one standard deviation.

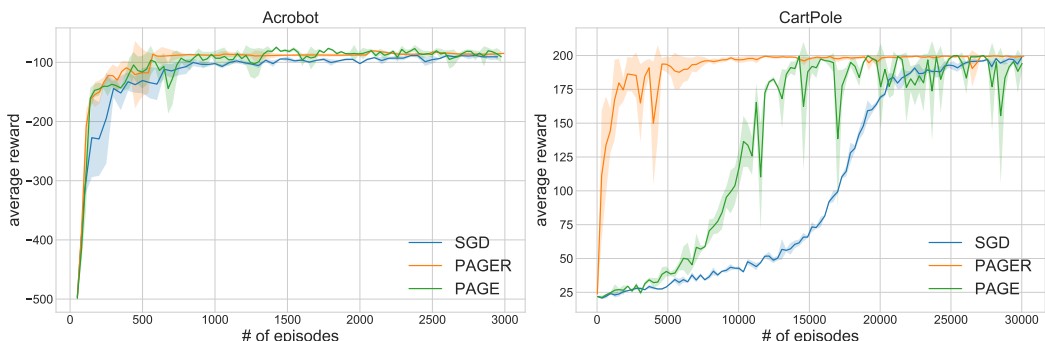

Figure 4: Performance of SGD, PAGER and PAGE on benchmark RL tasks.

**Results.** The empirical results shown in Figure 4 seem to be in line with our theoretical findings (Theorem 2). There are two interesting observations. First, SGD requires more time to converge compared to variance reduced methods. The difference is especially tangible for CartPole environment, where PAGER stabilizes at the maximal average reward 3 *times faster* than SGD. This is in line with the theoretical sample complexity gap between PAGER – $\mathcal{O}(\epsilon_f^{-2})$ and SGD – $\mathcal{O}(\epsilon_f^{-3})$. Second, PAGER converges much faster than its (non-restarted) variant PAGE on CartPole task, which shows empirically *the benefit of the restarting procedure*. Moreover, the behavior of PAGER is *more stable* near optimum. This observation is in accordance with the intuition described in Section C and our theoretical analysis because PAGER is able to reduce the variance term in (26) at the desired rate by varying parameters $p$ and $b$ over time.