# OpenReview forum: "Sharp Analysis of Stochastic Optimization under Global Kurdyka-Lojasiewicz Inequality"
_NeurIPS.cc/2022/Conference — NeurIPS 2022 Accept_

### Official Review · Reviewer_ekoT · 2022-07-09

**Rating:** 5
**Confidence:** 5
**Soundness:** 3 good
**Presentation:** 4 excellent
**Contribution:** 3 good

**Summary:**

The paper considers the stochastic optimization problem under the $\alpha$-PL condition and the order k expected smoothness condition. The paper establishes the finite-time convergence for the SGD with a restart algorithm. The result generalizes the previous analysis to a wider range of \alpha, and the sample complexity is later proven to be tight for restarting SGD. The paper then further improves the sample complexity by incorporating the variance reduction method PAGE into the restarting SGD algorithm.

**Questions:**

1. How generalized is Assumption 4 compared to the expected smoothness assumption [23,18,47]? Specifically, are there any applications with a concave h(t) other than $h(t)=t$? In addition, are there any applications where $\alpha \in (1,2)$?

2. The restart process in Algorithm 1 is important for establishing the sample complexity. Yet it seems there is only one condition on $N=KT$, while the explicit conditions on T and K, which captures the impact of the restarting process, cannot be found in the analysis of section 3.1. It would be great if the authors can explicitly discuss the choice of $T, K$.

3. The main results (Corollary 1, Proposition 1, and Theorem 2-3) hold for \alpha-PL functions. Why do the authors claim results hold under the more general KL-condition (e.g., in the title, in section 1.1)?

4. What are the theoretical challenges arising from incorporating PAGE into the stage-wise SGD? What is the techniques used to overcome the challenge? What is the reason for using PAGE instead of other variance-reduction methods?

Minor questions:

 There are typos in the paper:  ‘Assumption 7’ should be changed to ‘Assumption 6’, and there are question marks in line 620.


**Limitations:**

Yes.

**Strengths And Weaknesses:**

* Strengths:

The paper is well-structured and easy to understand. Section 3 offers an interesting analysis of the dynamics of stage-wise SGD under \alpha-PL condition. The work is sound and thorough—it contains a tight analysis of stage-wise SGD, and in addition, it also provides an analysis of the variance reduced variant and establishes an improved sample complexity for different \alpha.

* Weaknesses:

The theoretical generalization claimed by this paper lacks applications, e.g., the generalization of $\alpha=1 or 2$ to $\alpha \in [1,2]$ and the generalization of $h(t)=t$ to a concave $h(t)$ do not include more real-world applications. The extension of stage-wise SGD to PAGER lacks unique challenges therefore the improvement of sample complexity seems expected.

---

> ### Author Response · Authors · 2022-08-02
> **Question 1: Generality of Assumption 4**
>
> > 1. How generalized is Assumption 4 compared to the expected smoothness assumption [23,18,47]? Specifically, are there any applications with a concave h(t) other than h(t)=t? In addition, are there any applications where α∈(1,2)?
>
> We appreciate the insightful comment. As we discussed in the paragraph after this assumption, it is more general than the expected smoothness (ES). More precisely, when $h(t)=t$ and costs are one $b_k=1$ we have ES. As discussed by [KR21], in subsampling schemes or in compression schemes, we have assumption 4 with $h(t)=t$. It is important to emphasize that for instance the appearance of $h(t)=t$ in these schemes is a result of an additional smoothness assumption on the subsampled functions (Assumption 3 in [NR21]).  In general, it is true that most common applications satisfy ES with h(t)=t, but this by no means would limit our contribution. It is noteworthy that our framework allows the researcher from different fields to handle a wider set of non-convex functions and stochastic oracles.
>
> Regarding applications with $1<\alpha<2$, we have not yet seen any real world application but again this does not limit our contribution. Nonetheless, to emphasize that such functions exist, we present an example. Please see Example 5 in Appendix A.1.
> $f(x)= x^{\frac{\alpha}{\alpha-1}}\Big(\sin^2(x^{-\frac{1}{\alpha-1}}) + 1\Big) + g(x)$,
> Where $g'(x)=\sin(2x^{-\frac{1}{\alpha-1}})/(\alpha-1)$ and $g(0)=0$.
> This function is L-smooth, non-convex, and satisfies the PL condition for $1<\alpha<2$.
>
> [KR21] Ahmed Khaled and Peter Richtárik. Better theory for SGD in the nonconvex world.

---

> ### Author Response · Authors · 2022-08-02
> **Question 2: Restart process in SGD**
>
> > 2. The restart process in Algorithm 1 is important for establishing the sample complexity. Yet it seems there is only one condition on N=KT, while the explicit conditions on T and K, which captures the impact of the restarting process, cannot be found in the analysis of section 3.1. It would be great if the authors can explicitly discuss the choice of T,K.
>
> The results of Section 3 are under a fixed $T$, i.e., unlike PAGER,  $T=\Omega(1/\min_j \omega_j)$ remains unchanged for different $k$. We have elaborated on this in the new version, Theorem 1.

---

> ### Author Response · Authors · 2022-08-02
> **Question 3: Result for general KL condition**
>
> > 3. The main results (Corollary 1, Proposition 1, and Theorem 2-3) hold for $\alpha$-PL functions. Why do the authors claim results hold under the more general KL-condition (e.g., in the title, in section 1.1)?
>
> One of the main contributions of our work is to introduce a general framework for analyzing the convergence rate of SGD with objectives that satisfy the global KL condition (resp. Theorem 1). However, this general KL condition relies on the function $\phi(t)$. Without specifying the form of $\phi(t)$, deriving  the convergence rate of SGD is quite challenging. But as soon as the form of this function is given (e.g., $\phi(t)=t^{1/\alpha}$ that is the case for functions satisfying the PL condition),  then our framework is able to find the convergence rate (resp. Corollary 1).
>
> As additional examples, here we present the convergence rate of SGD under the global KL assumption (Assumption 2) with
> $\phi(t) = \sqrt{t \ln(t+1)}$ and $h(t) = \ln(1+t)$,
> and
> $\phi(t) = \min\{t, \sqrt{t}\}$ and $h(t) = t$. Note that this  setting occurs in machine learning applications when the loss function is squared cross entropy [Kevin et al.'22]. We have also mentioned this in the revised version page 4.
> We have discussed both cases in detail in Appendix A.2 of the revised version.
>
> In the first case, it is straightforward to see that the stationary point (i.e., solution to equation (8)) is $r(\eta)=\Theta(\eta^{1/2})$ which means $\nu=0.5$ in Theorem 1. Moreover, the parameter $\zeta$ in Theorem 1 will be one. Hence, this Theorem implies that $\delta_k=O(k^{-0.5})$ or equivalently the convergence rate of SGD will be of the order $O(\epsilon^{-2})$. Similarly, we can obtain the convergence rate for the second case as $O(\epsilon^{-3})$.
>
> [Kevin et al.'22] Scaman, Kevin, et al.. "Convergence Rates of Non-Convex Stochastic Gradient Descent Under a Generic Lojasiewicz Condition and Local Smoothness." International Conference on Machine Learning. PMLR, 2022.

---

> ### Author Response · Authors · 2022-08-02
> **Question 4: Theoretical challenges of analyzing variance reduction**
>
> > The extension of stage-wise SGD to PAGER lacks unique challenges therefore the improvement of sample complexity seems expected.
>
> > 4. What are the theoretical challenges arising from incorporating PAGE into the stage-wise SGD? What is the techniques used to overcome the challenge? What is the reason for using PAGE instead of other variance-reduction methods?
>
> We thank the reviewer for the valuable questions, we provided additional discussion about why restarting is important in the revised version in Appendix C. The stage-wise strategy is crucial for obtaining $\varepsilon^{-\frac{2}{\alpha}}$ sample complexity of PAGER. For instance, we demonstrate in Appendix C that merely applying PAGE with constant probability $p$ will not work. Our solution to this problem is a modified algorithm – PAGER, which varies its parameters in a stage-wise manner.
> We highlight that PAGE is known to be **optimal** in the general non-convex case in terms of both finite sum and pure stochastic case. The optimality is achieved simultaneously for both $\epsilon$ and $n$. Moreover, PAGE achieves the best known complexity for $2$-PL case [Li et al, 2021]. Notice that other popular variance reduced methods such as SAGA, SAG, SVRG have worse dependence on $n$ for general nonconvex and $2$-PL objectives [Reddi et al, 2016]. This observation serves as the main motivation to extend PAGE-like method to $2$-PL functions because one may expect similar optimal/SOTA results. Indeed, it is the case! As we elaborate in Section 4.2, in the most interesting case $\alpha = 1$, PAGER is optimal because it achieves $O(1/\epsilon^{-2} )$ sample complexity matching the lower complexity bound for stochastic convex optimization. Of course, it is expected that the relatives of PAGE such as SARAH [Nguyen et al, 2019], SPIDER [Fang et al, 2018] could be possibly also analyzed under $\alpha$-PL. However, we expect that the analysis will be involved due to their double loop structure.
>
> Zhize Li, Hongyan Bao, Xiangliang Zhang, Peter Richtarik. PAGE: A Simple and Optimal Probabilistic Gradient Estimator for Nonconvex Optimization. Proceedings of the 38th International Conference on Machine Learning, 2021.
>
> Sashank J. Reddi, Ahmed Hefny, Suvrit Sra Barnabas, Poczos Alex Smola. Stochastic Variance Reduction for Nonconvex Optimization. JMLR. 2016.
>
> Lam M. Nguyen, Marten van Dijk, Dzung T. Phan, Phuong Ha Nguyen, Tsui-Wei Weng, and Jayant R. Kalagnanam. Finite-Sum Smooth Optimization with SARAH. arXiv preprint arXiv:1901.07648, 2019.
>
> Cong Fang, Chris Junchi Li, Zhouchen Lin, and Tong Zhang. SPIDER: Near-Optimal Non- Convex Optimization via Stochastic Path-Integrated Differential Estimator. In Advances in Neural Information Processing Systems, volume 31, 2018.

---

> ### Author Response · Authors · 2022-08-02
> **Minor questions.**
>
> > There are typos in the paper: ‘Assumption 7’ should be changed to ‘Assumption 6’, and there are question marks in line 620.
>
> Thank you for pointing out these typos. We have fixed them.

---

> ### Author Response · Authors · 2022-08-08
> **Author-Reviewer discussion period**
>
> Dear Reviewer ekoT,
> As the discussion period is ending today, we appreciate if the reviewer could take a chance to read our responses. Have we addressed your main questions about the generality of the assumptions and technical challenge?  Do you have any further questions that we can clarify?

---

### Official Review · Reviewer_ws9Y · 2022-07-11

**Rating:** 7
**Confidence:** 3
**Soundness:** 3 good
**Presentation:** 3 good
**Contribution:** 3 good

**Summary:**

This paper studies the convergence of SGD and PAGE under smoothness, the Kurdyka-Lojasiewicz inequality, and the expected smoothness assumption. The authors show that the iterates generated by SGD follow a dynamic (Lemma 1) that can be tuned to ensure the optimal convergence rate using a certain step size scheme (Theorem 1 and Corollary 1). The authors show that using (with-replacement) mini-batching in the online setting can not improve the convergence rate of SGD further, and that their solution of the SGD dynamic they start with is optimal, although whether the dynamic they start with is optimal is not known. By extending the variance-reduced PAGE algorithm to the KL setting, the authors obtain improved convergence rates for online as well as finite-sum problems.


**Questions:**

- Can your result be used to handle other forms of minibatching as in [1]?
- What about the dependence on the condition number? It is hard to parse and compare how your condition number dependence differs from prior work.
- Lemma 4 seems to use a similar step size scheme to [2], can you discuss the similarities/differences?



Post-rebuttal:
I thank the authors for their clarifying remarks, I have raised my score as the response is satisfactory.

[1] Robert Mansel Gower, Nicolas Loizou, Xun Qian, Alibek Sailanbayev, Egor Shulgin, Peter Richtárik. SGD: General Analysis and Improved Rates, PMLR 97 (2019).
[2] Sebastian U. Stich Unified Optimal Analysis of the (Stochastic) Gradient Method, arXiv:1907.04232 (2019).


**Limitations:**

See the previous sections for limitations. Potential negative societal impacts are not applicable here.


**Strengths And Weaknesses:**

Main points:
- (Strength) The setting studied (convergence under KL and Assumption 4) is very general, it covers several interesting applications that are not covered by existing analysis, and extends the applicability of results on the convergence of SGD and PAGE to several new applications (e.g. those mentioned in lines 44-46).
- (Strength) The rates provided by the paper (in Corr. 1 and Thm. 3) recover the best prior rates when specialized to the setting of \(\alpha = 2\), and extend these settings for general \(\alpha\) and for SGD, under assumption 4 specifically. This means we gain more generality at no additional cost. Furthermore, generalizing the proofs is not straightforward from the prior work.
- (Weakness) Sec. 3.3 talks about tightness, but in my opinion it is very misleading. The notion of tightness used here is the tightness of solving the particular recurrence the authors get in (eq. 7); But the result of Lemma 1 may not itself be tight at all.
- (Weakness) While Sec 3.2 shows that minibatching cannot improve the convergence of SGD, this is only for minibatching with-replacement in the online setting. There are other forms of minibatching that can potentially improve the convergence rate of SGD, see e.g. [1].

Overall: I think that while the paper has some drawbacks, it presents a nice contribution to the literature on SGD and PL. Therefore, I recommend acceptance.

[1] Robert Mansel Gower, Nicolas Loizou, Xun Qian, Alibek Sailanbayev, Egor Shulgin, Peter Richtárik. SGD: General Analysis and Improved Rates, PMLR 97 (2019).

---

> ### Author Response · Authors · 2022-08-02
> **Weaknesses 1-2: Tightness and mini-batching**
>
> > (Weakness) Sec. 3.3 talks about tightness, but in my opinion it is very misleading. The notion of tightness used here is the tightness of solving the particular recurrence the authors get in (eq. 7); But the result of Lemma 1 may not itself be tight at all.
>
> This is a valid observation. As we clearly stated our tightness result is only valid when the recursion in (7) is an equality instead of inequality. Herein, we do not provide an information-theoretic lower complexity bound for stochastic first-order algorithms under KL assumption, and we do not claim that we do. Note that the tightness result in Proposition 1 (Proposition 2 in the revision) only refers to the rate of convergence of a particular recursion (under any choices of stepsizes), not an algorithmic class. In fact, it seems very challenging to establish such lower bounds even for a specific stochastic algorithm. We are not aware of any results on this topic
>
> > (Weakness) While Sec 3.2 shows that minibatching cannot improve the convergence of SGD, this is only for minibatching with-replacement in the online setting. There are other forms of minibatching that can potentially improve the convergence rate of SGD, see e.g. [1].
>
> We thank the reviewer for the comment. Yes, indeed. In this work, we only refer to mini-batching with-replacement. We have moved our discussion about why this type of mini-batching does not improve SGD’s complexity to Appendix C in the revised version.

---

> ### Author Response · Authors · 2022-08-02
> **Questions: 1-3: other forms of mini-batching, dependence on $\kappa$, step-sizes**
>
> > 1. Can your result be used to handle other forms of minibatching as in [1]?
>
> This is an interesting question. Although it requires precise consideration, a quick analysis shows that we can apply our method to handle several forms of sampling such as independent sampling, single element sampling, and most like the $\tau$-nice sampling and partition sampling, for precise definitions please see [1]. The high-level explanation is that these forms of sampling provide us an estimator of the gradient with their specific constants $A$, $B$, $C$, and the cost $b_k$ (see Assumption 4 in our work). As long as we can compute the constants and the cost, we are able to apply our results. Such constants were computed for different sampling strategies in [1] and one can use them in order to obtain the rates.
> [1] Ahmed Khaled and Peter Richtárik. Better theory for SGD in the nonconvex world.
>
> > 2. What about the dependence on the condition number? It is hard to parse and compare how your condition number dependence differs from prior work.
>
> We showcase the dependence on the condition number in Table 1. It turns out that for SGD, the dependence is the same as for prior work under BV assumption. Notice that the dependence on $\kappa$ is linear both in the stochastic and finite-sum setting and it is unclear whether this can be further accelerated to $\sqrt{\kappa}$ for $2$-PL functions. We highlight that even in the deterministic settings, this remains an open question [Danilova et al, 2021].
> Marina Danilova, Grigory Malinovsky “Averaged Heavy-Ball Method”, 2021.
>
> > 3. Lemma 4 seems to use a similar step size scheme to [2], can you discuss the similarities/differences?
>
> This is a correct observation. In fact, Lemma 4 is a generalization of the scheme in [2]. We have emphasized the similarities of their schemes in Appendix B.5 before Lemma 3. Note that Lemma 4 is now changed to Lemma 3 in the new version.
>
> > [1] Robert Mansel Gower, Nicolas Loizou, Xun Qian, Alibek Sailanbayev, Egor Shulgin, Peter Richtárik. SGD: General Analysis and Improved Rates, PMLR 97 (2019).
> > [2] Sebastian U. Stich Unified Optimal Analysis of the (Stochastic) Gradient Method, arXiv:1907.04232 (2019).

---

### Official Review · Reviewer_cXuP · 2022-07-12

**Rating:** 6
**Confidence:** 4
**Soundness:** 2 fair
**Presentation:** 2 fair
**Contribution:** 2 fair

**Summary:**

The paper provides rates of convergence under various assumptions for global KL functions - which is rich enough to include many popular loss functions in practice. The authors provide rates for SGD algorithm, and then provide a new algorithm called PAGER which enjoys improved rates due to variance reduction. They then show how to apply this algorithm for finite sum optimization problems.


**Questions:**

Please refer to other sections!

**Limitations:**

No foreseeable negative societal impacts.

**Strengths And Weaknesses:**

Strengths: The class of global KL is fairly rich and stochastic optimization is not well understood in this class.

Weaknesses: I found the contributions of the paper to be a bit underwhelming and lacking novelty. Let me explain:

1. The upper bound provided in Corollary 1 and the lower bound in Proposition 1 do not fit well with each other. The lower bound only holds for a particular step-size sequence and does not say much about the minimax rates for SGD for this class.

2. Given the above, I am not sure how valuable is the contribution of improved rates under variance reduction.

3. Lemma 2 is vacuous as it seems to be based on just comparing upper bounds which may not be tight.


4. The theoretical ideas used in the SGD analysis, variance reduction, etc are not novel. In the end, the paper seems to be combining various previously known technical ideas, without any new high-level algorithmic insights. The paper seems to be missing an important reference: Second-Order Information in Non-Convex Stochastic Optimization: Power and Limitations, Arjevani et. al. 2020.

5. There are still many gaps in the corresponding rates for KL functions.

---

> ### Author Response · Authors · 2022-08-02
> **Weaknesses 1-3: I found the contributions of the paper to be a bit underwhelming and lacking novelty**
>
> > Weaknesses: I found the contributions of the paper to be a bit underwhelming and lacking novelty. Let me explain:
>
> > 1. The upper bound provided in Corollary 1 and the lower bound in Proposition 1 do not fit well with each other. The lower bound only holds for a particular step-size sequence and does not say much about the minimax rates for SGD for this class.
>
>
> We explain in the beginning of Appendix C why we believe that the given rate of SGD may not be improved. Consider the special case $\alpha = 1$, the main intuition is that in order to achieve a descent property (with constant step-size), one needs to use $O(1/\epsilon^{-2})$ number of samples. This results in $O(1/\epsilon^{-3})$ total sample complexity of the algorithm because deterministic gradient descent has $O(1/\epsilon)$ iteration complexity. However, in our work, we do not provide an information-theoretic lower complexity bound for stochastic first-order algorithms under KL assumption, and we do not claim that we do. Note that the tightness result in Proposition 1 (Proposition 2 in the revision) only refers to the rate of convergence of a particular recursion (under any choices of stepsizes), not an algorithmic class. In fact, it seems very challenging to establish such lower bounds even for a specific stochastic algorithm. We are not aware of any results on this topic even for the well studied 2-PL case (except for those which are based on strongly convex functions), but we believe this is an interesting research direction.
>
> > 2. Given the above, I am not sure how valuable is the contribution of improved rates under variance reduction.
>
> We would like to emphasize that based on Corollary 1 and Proposition 1, the rate of SGD cannot be improved (assuming that recursion (7) is an equality). However, in Section 4, we show that by applying variance reduction, we can achieve better rate and in particular for 1-PL functions that are important in RL applications, our algorithm PAGER achieves the optimal convergence rate that is $O(\epsilon^{-2})$. It is noteworthy that no known algorithm has so far achieved $O(\epsilon^{-2})$ rate for 1-PL functions and for 2-PL functions, variance reduction does not help improving rates for the online setting.
>
> > 3. Lemma 2 is vacuous as it seems to be based on just comparing upper bounds which may not be tight.
>
> Thanks for pointing it out. We agree with the reviewer that the statement of Lemma 2 may be confusing. We have now replaced this statement with Proposition 1. While Corollary 1 focuses on iteration complexity, the goal of Proposition 1 is to show that under Assumption 4, increasing the cost of estimator $b_k$ does not improve the total sample complexity.

---

> ### Author Response · Authors · 2022-08-02
> **Weaknesses 4-5: The theoretical ideas used in the SGD analysis, variance reduction, etc are not novel.**
>
> > 4. The theoretical ideas used in the SGD analysis, variance reduction, etc are not novel. In the end, the paper seems to be combining various previously known technical ideas, without any new high-level algorithmic insights. The paper seems to be missing an important reference: Second-Order Information in Non-Convex Stochastic Optimization: Power and Limitations, Arjevani et. al. 2020.
>
> We respectfully disagree with the reviewer about the concerns on the novelty of the ideas. To the best of our knowledge the proposed proof technique of convergence of SGD under KL condition (based on Lemma 1 and Theorem 1) was not discussed in the literature before and are useful in analyzing convergence of global KL functions that are not covered by PL functions. We highlight that the analysis of variance reduction under KL ($\alpha$-PL) condition is new and contains both new algorithmic ideas (parameter restart) and challenges in the analysis. Moreover, we obtain for the first time the optimal $O(\epsilon_{f}^{-2})$ complexity for 1-PL functions using the 1st order method, which itself is a valuable contribution to the field.
> We thank the reviewer for pointing out the paper [Arjevani et. al. 2020], which discussed the analysis and limits of 2-order methods for general non-convex optimization. While we believe our focus is quite different, we will cite the paper in our revision as an orthogonal technique to variance reduction.
>
> > 5. There are still many gaps in the corresponding rates for KL functions.
>
> We appreciate it if the reviewer can further elaborate which gaps they have in mind regarding the rates for KL functions. We are happy to engage in more discussions with the reviewer if the answer is not covered by our clarifications above.

---

> ### Author Response · Authors · 2022-08-08
> **Author-Reviewer discussion period**
>
> Dear Reviewer cXuP,
> As the discussion period is ending soon today, we appreciate if the reviewer could take a chance to read our responses. Have we addressed your main concerns on the novelty?  Do you have any further questions that we can clarify?

---

### Official Review · Reviewer_FAsV · 2022-07-13

**Rating:** 4
**Confidence:** 3
**Soundness:** 2 fair
**Presentation:** 2 fair
**Contribution:** 2 fair

**Summary:**

This paper studies the complexity of the first-order algorithms with being able to find the global optimal solution of nonconvex objective functions which satisfy the KL condition. This work uses a dynamic system to characterize the evolution of the iterates generated by the proposed algorithms. It is shown in this paper that the derived convergence rate of the variants of SGD is tight and further a variance reduced algorithm is also proposed with a convergence rate of matching the lower bound.

**Questions:**

1, it is quite confusing that why $g_k$ needs to be dependent on $k$? how does $K$ take the role in the convergence rate?

2, in theorem 1, how many $\omega_j$s? it seems that there is only one $\omega_k$ show in eq. 9. How about others?

3, in theorem 1, whether $\nu$ is also nonnegative?

4, in theorem 1, when $\zeta$ is large, then $\delta_k$ shrinks fast but $\eta_k$ is small, which contradicts the intuition that a large step size gives a fast convergence rate.

5, line 230, when $\beta=0$, [14] gives the result but corollary 1 requests $\beta\in(0,1]$ which means that $\beta$ cannot be 0 or corollary does not cover the existing results.

6, I agree that the convergence result built upon the last iterate makes more sense, but how does the stochasticity of the gradient estimate take place in the final convergence result?

7, what is the main reason of using the restart strategy? how to compare the step size $\eta$ with the classic one in SGD? is this choice of the learning rate critical to show the convergence?

8, line 297, how is $D$ obtained in the analysis? is it dependent on $\mu$ or $L$?

9, in theorem 3, why is the step size $\eta_t$ dependent on $n$? when $n$ is large, the learning rate will be extremely small, right?

10, why did not include some numerical results?



**Limitations:**

The limitations of this work are not mentioned, but I think it is fine as it is a truly theoretical one of discussing some math properties.


**Strengths And Weaknesses:**

Strengths:
This paper studies a recent popularized question of showing the convergence rate of the first-order algorithms to the global optimal solutions of the nonconvex objective functions under the KL condition. This class of problems indeed covers a wide range of problems in machine learning. Especially, it is good to see that this work has already listed several motivating examples whose objective functions satisfy the KL condition.

More importantly, this paper provides the convergence rate of a variant of the SGD algorithm in this setting and proposes a new algorithm that can achieve the improved rate to find the global optimal solution. Theorem 1. is general, which gives the iteration complexity of the SGD with restarts under the KL condition.

Weaknesses:
Even though I believe that this is a truly interesting work and studying a worth well optimization problem, the main concerns are the quality of the presentation, significance of the theoretical results, and numerical justification of the algorithms. To be more specific, please see below.

1, The presentation of this work needs more attention. A thorough comparison between this work and existing ones should be addressed in multiple dimensions, e.g., learning rates, hyper-parameters if any, complexity of the algorithm. Also there are other issues. See the next session.

2,  Although the convergence rate achieved by the SGD with restarts is applicable for general case (KL), the rate matches the existing ones for the PL case which is the most practical setting for applications. How this extension is important to the machine learning problems is not clearly discussed or what are the challenges of showing the convergence beyond PL are not particularly discussed. Note that some discussion included in page 6 mainly focuses on the generality of the assumption 4 rather than KL. So, which assumption or condition is the truly restrict/general one that needs to be conveyed in this work is not clear.

3, it is not surprising that PAGER can achieve a better convergence as the variance reduction has been applied. In practice, VR based algorithm would not perform well in terms of the generalization error. Also, the theoretical results are only applicable to the PL case rather than the KL case, but I presumably the main topic of this work is about showing the iteration complexity of the algorithms under the KL condition.

4, Both alg.1 and alg.2 are variants of the existing ones. It is necessary to showcase the numerical performance of these algorithms with a comparison with the baselines.


============================= After rebuttal ====================

Thanks for the authors' detailed response. I have increased the score. The concern regarding the small step size remains, which is not about the correctness of the proof but a concern about the dependence of the step size on the hyper-parameters, e.g., $n$.

---

> ### Author Response · Authors · 2022-08-02
> **Presentation of the work**
>
> > 1, The presentation of this work needs more attention.
>
> We thank the reviewer for the suggestions on improving the presentation. We have addressed each suggestion/concern separately and made changes in the revision where we believe it is appropriate.

---

> ### Author Response · Authors · 2022-08-02
> **Comparison with other work**
>
> > A thorough comparison between this work and existing ones should be addressed in multiple dimensions, e.g., learning rates, hyper-parameters if any, complexity of the algorithm.
>
> **Learning rates and complexity.**
>
> In Remark 1, we compare SGD with related works in terms of sampling complexity.  Only a few works [Fontaine et. al., 21], [Yuan et. al., 21]  considered stochastic setting under $\alpha$-PL (\alpha < 2). Regarding general KL, we are not aware of any works in the stochastic case. [Yuan et. al., 21] focus on $1$-PL condition and obtain $\widetilde{O}(\epsilon^{-3})$ sample complexity (Theorem C.2). We also obtain $\epsilon^{-3}$ sample complexity (removing the logarithmic factor). While it is not straightforward to directly compare to the learning rates in [Yuan et. al., 21], we notice that they are of the same order (as in our Corollary 1) $O(k^{-\frac{2}{3}})$ after k iterations. [Fontaine et. al., 21] study $\alpha$-PL condition and also obtain  $\epsilon^{-\frac{4-\alpha}{\alpha}}$ sample complexity. Regarding the learning rates, it turns out that even though we use a different proof technique, we recover the same learning rates as in [Fontaine et. al., 21]. For instance, if we set $T=1$ and $b_k=1$, we get $\eta_k = O(k^{-\frac{2}{4-\alpha}})$. We have added a comment about this in the revised version.
>
> **Hyper-parameters.**
>
> Note that SGD essentially does not have hyper-parameters (the stage-length in restarted SGD can be always set to $T=1$ and does not make a difference, please refer to our explanation in the reply to question 7).
>
> [Fontaine et. al., 21] Xavier Fontaine, Valentin De Bortoli, and Alain Durmus. Convergence rates and approximation results for SGD and its continuous-time counterpart. PMLR, 2021.
>
> [Yuan et. al., 21] Rui Yuan, Robert M Gower, and Alessandro Lazaric. A general sample complexity analysis of vanilla policy gradient. arXiv preprint arXiv:2107.11433, 2021.

---

> ### Author Response · Authors · 2022-08-02
> **Discussion of KL functions beyond $\alpha$-PL**
>
> > 2, Although the convergence rate achieved by the SGD with restarts is applicable for general case (KL), the rate matches the existing ones for the PL case which is the most practical setting for applications. How this extension is important to the machine learning problems is not clearly discussed or what are the challenges of showing the convergence beyond PL are not particularly discussed. Note that some discussion included in page 6 mainly focuses on the generality of the assumption 4 rather than KL. So, which assumption or condition is the truly restrict/general one that needs to be conveyed in this work is not clear.
>
>
> PL functions are special cases of KL functions. There are scenarios in machine learning applications in which the objective function is no longer PL. For example, consider the loss function of the form squared cross entropy [Kevin et al.'22]. Hene, it is important to have an understanding of the convergence rate of SGD for such classes of functions.
> The main challenge of extending the SGD analysis from PL to KL functions is that different functions satisfy the KL condition with different $\phi(t)$ functions. Therefore, without specifying the form of $\phi(t)$ it is quite challenging to obtain the precise convergence rate of SGD. However, in our work, we introduced a general framework that  as soon as the form of this function is known (e.g., $\phi(t)=t^{1/\alpha}$ for functions satisfying the PL condition) then we are able to specify the convergence rate (e.g., corollary 1).
>
> As additional examples,  we also present the convergence rate of SGD under the global KL assumption (Assumption 2) with
> $\phi(t) = \sqrt{t \ln(t+1)}$ and $h(t) = \ln(1+t)$,
> and
> $\phi(t) = \min\{t, \sqrt{t}\}$ and $h(t) = t$. Note that this a setting that occurs in machine learning application when the loss function is squared cross entropy [Kevin et al.'22].
>
> We have discussed both cases in detail in Appendix A.2 of the revised version.
> In the first case, it is straightforward to see that the stationary point (i.e., solution to equation (8)) is $r(\eta)=\Theta(\eta^{1/2})$ which means $\nu=0.5$ in Theorem 1. Moreover, the parameter $\zeta$ in Theorem 1 will be one. Hence, this Theorem implies that $\delta_k=O(k^{-0.5})$ or equivalently the convergence rate of SGD will be of the order $O(\epsilon^{-2})$. Similarly, we can obtain the convergence rate for the second case as $O(\epsilon^{-3})$.
>
> [Kevin et al.'22] Scaman, Kevin, et al.. "Convergence Rates of Non-Convex Stochastic Gradient Descent Under a Generic Lojasiewicz Condition and Local Smoothness." International Conference on Machine Learning. PMLR, 2022.

---

> ### Author Response · Authors · 2022-08-02
> **"Not surprising that PAGER can achieve a better convergence" and "generalization error"**
>
> > 3, it is not surprising that PAGER can achieve a better convergence as the variance reduction has been applied. In practice, VR based algorithm would not perform well in terms of the generalization error. Also, the theoretical results are only applicable to the PL case rather than the KL case, but I presumably the main topic of this work is about showing the iteration complexity of the algorithms under the KL condition.
>
> We believe there might be a misunderstanding about what we achieve in this work.
>
> $\bullet$ We highlight that indeed, variance reduction techniques such as PAGE were proven to achieve better convergence for stochastic nonconvex problems, but **for special class of nonconvex functions** such as 2-PL functions (or strong convex as a special case), **they cannot improve over SGD**. It remains elusive before this paper whether the variance reduction technique is useful for $\alpha$-PL function. We provide an **informative answer to this open question**.
>
> $\bullet$ The analysis of any variance reduction scheme under KL ($\alpha$-PL), except for the case $\alpha = 2$, **is not straightforward**. For instance, we demonstrate in Appendix C that merely applying PAGE with constant probability $p$ will not work. Our solution to this problem is a modified algorithm – PAGER, which varies its parameters in a stage-wise manner. We highlight that several works previously studied variance reduction schemes under KL ($\alpha$-PL) condition [Hu et al, 2021] , [Li et al, 2017], however, even in a simpler finite sum case, their results are not satisfactory. The analysis in [Hu et al, 2021]  is only asymptotic analysis and the dependence on the parameters $\kappa$ and $n$, which are important in practice for quantifying the improvement over GD  and SGD, are ignored. [Li et al, 2017] propose SVRG based algorithm, however, they do not provide theoretical analysis except for the well studied case $\alpha = 2$.
>
> $\bullet$ **Generalization error** of variance reduced methods. The lack of generalization of VR  is primarily reported for purely general nonconvex setting or training deep learning [Defazio et al, 2018]. This is because only convergence to stationary points is guaranteed (and stationary points may not generalize well).  However, for the global KL ($\alpha$-PL) function, VR methods are guaranteed to converge to global optimal solutions and the situation is completely different.
>
> [Hu et al, 2021] Jia Hu, Congying Han, Tiande Guo, and Tong Zhao. On the Convergence of Stochastic Splitting Methods for Nonsmooth Nonconvex Optimization. 2021.
>
> [Li et al, 2017] Qunwei Li, Yi Zhou, Yingbin Liang, and Pramod K. Varshney. Convergence analysis of proximal gradient with momentum for nonconvex optimization. In Proceedings of the 34th International Conference on Machine Learning, Proceedings of Machine Learning Research, 2017.
>
> [Defazio et al, 2018 Aaron Defazio, Léon Bottou. On the Ineffectiveness of Variance Reduced Optimization for Deep Learning. 2018.

---

> ### Author Response · Authors · 2022-08-02
> **Numerical experiments**
>
> > 4, Both alg.1 and alg.2 are variants of the existing ones. It is necessary to showcase the numerical performance of these algorithms with a comparison with the baselines.
>
> We provide numerical simulations on RL problems in the revision. The empirical results shown in Figure 4 (Appendix E) align with our theoretical findings Theorem 2. There are two interesting observations. First, SGD requires more time to converge compared to variance reduced methods. The difference is especially tangible for CartPole environment, where PAGER stabilizes at the maximal average reward $3$ times faster than SGD. This is in line with the theoretical sample complexity gap between PAGER -- $O(\epsilon_f^{-2})$ and SGD -- $O(\epsilon_f^{-3})$. Second, PAGER converges much faster than its (non-restarted) variant PAGE on CartPole task, which shows empirically \textit{the benefit of the restarting procedure}. Moreover, the behavior of PAGER is more stable near optimum. This observation is in accordance with the intuition described in Section C and our theoretical analysis because PAGER is able to reduce the variance term at the desired rate by varying parameters $p$ and $b$ over time.

---

> ### Author Response · Authors · 2022-08-02
> **Questions 1-6**
>
> > 1, it is quite confusing that why gk needs to be dependent on k? how does K take the role in the convergence rate?
> $g_k$ is an unbiased estimation of the gradient $\nabla f$ at the k-th iteration. Therefore, in general it may depend on the iteration index k.
>
> K denotes the total number of iterations and it determines when we have achieved an $\epsilon$-stationary point. There is a straightforward relation between the convergence rate in terms of $\epsilon$ and in terms of $K$. We have mentioned this in the paper.
>
> > 2, in theorem 1, how many ωjs? it seems that there is only one ωk show in eq. 9. How about others?
>
> In Equation (9), each iteration (k) has its own $\omega$. Hence, we require a set of $\omega$s which we denote by $\{\omega_j\}_{j\geq0}$.
>
> > 3, in theorem 1, whether ν is also nonnegative?
>
> $\nu$ is also nonnegative. We have clarified this in the paper.
>
> > 4, in theorem 1, when ζ is large, then δk shrinks fast but ηk is small, which contradicts the intuition that a large step size gives a fast convergence rate.
>
> This theorem states that if $\nu$ and $\zeta$ exist such that Equation (9) holds then  $\delta_k$ has the rate of $O(k^{-\nu\zeta})$. In high-level, this is because the recursion in (7) has a stationary point denoted by $r(\eta)$ that is not necessarily placed at the origin (i.e., 0). On the other hand, by shrinking $\eta$, we can move this stationary point toward zero. But there is a restriction. That is, by decreasing $\eta$, the stationary point of the recursion will change and thus it requires some iterations to converge (or  get close enough) to this stationary point. If the step-sizes are selected too big, then the recursion cannot converge to the stationary point in a fixed small number of iterations which may result in divergence of the overall SGD. Therefore, the intuition that a larger step-size necessarily gives better convergence rate is not accurate.
>
> > 5, line 230, when β=0, [14] gives the result but corollary 1 requests β∈(0,1] which means that β  cannot be 0 or corollary does not cover the existing results.
>
> Thank you for this comment. It is important to realize that when $\beta=0$, then it is similar to having a constant $2A$ on the right hand side of equation (6) which is equivalent to having a bounded variance assumption. As we also mentioned in the last paragraph of page 4, this is a special case of our assumption when $h(t)=0$ and $\tau=0$. We will clarify this in the paper that $\beta=0$ is in fact equivalent to $h(t)=0$ and $\tau=0$. We have clarified this is in the revised version.
>
> > 6, I agree that the convergence result built upon the last iterate makes more sense, but how does the stochasticity of the gradient estimate take place in the final convergence result?
>
> In Table 1, we showcase the dependence on constants  $\sigma^2$, $\kappa$ and $\mu$ under bounded variance assumption. We thank the reviewer for this comment, in fact, we took a closer look at this and fixed a typo related to this in the revision, which was in the dependency in $\kappa$. It turns out that the dependency for PAGER is $\kappa^2$, but it enters additively with $\sigma^2$. While for SGD $\kappa$ is multiplied with $\sigma^2$ in the complexity.

---

> > ### Comment · Reviewer_FAsV · 2022-08-09
> > **Confusion about moving the stationary point**
> >
> > I am really confused by "we can move this stationary point toward zero" and " the stationary point of the recursion will change". The stationary points of the problem is fixed, which is only dependent on the problem rather than algorithms. How do we move it or change it?

---

> > > ### Author Response · Authors · 2022-08-09
> > > **About moving the stationary point**
> > >
> > > We thank the reviewer for the question. To clarify this matter, it is important to notice that the stationary point is a function of the step size $\eta$ and as $\eta$ goes to zero, the stationary point moves toward zero. Hence, by choosing the step-sizes wisely, we can move the stationary point to zero. We hope this addresses your question.

---

> ### Author Response · Authors · 2022-08-02
> **Questions 7-10**
>
> > 7, what is the main reason of using the restart strategy? how to compare the step size η with the classic one in SGD? is this choice of the learning rate critical to show the convergence?
>
> There are two main motivations to consider restarting strategy for SGD. The first motivation is practical. Notice that the step-size schedule in the restarted SGD is essentially a popular step-decay (piecewise constant) strategy. Such technique is commonly used in practice and often outperforms the classical diminishing step-size schedule for SGD [Krizhevsky et al, 2012], [He et al, 2016]. Therefore, we believe it is important to provide solid theoretical foundations for such piecewise constant step-size strategy.
> The second motivation comes from the analysis perspective. Notice that the restart strategy is crucial for obtaining $\varepsilon^{-\frac{2}{\alpha}}$ sample complexity of PAGER. We provide discussion about why restarting is important and the intuition why SGD is not sufficient in Appendix C. Therefore, it becomes interesting to find out if restarts help to improve the sample complexity of SGD (as it is the case for variance reduced method).
> Notice that setting $T=1$, our restarted SGD reduces to the standard SGD and the correspondence for the classical step-size becomes evident. Here, we analyze a more flexible step-size schedule, than a classic one (allowing to set arbitrary $T\geq 1$).
> Overall, we did our best to extensively analyze SGD from different angles (using minibatch, restarting, constant/varying step-sizes); however, our conclusion is that at least following this type of analysis, the improvement over $\varepsilon^{-\frac{4-\alpha}{\alpha}}$ cannot be obtained.
>
> A. Krizhevsky, I. Sutskever, and G. E. Hinton. Imagenet classification with deep convolutional neural networks. In Advances in Neural Information Processing Systems, 2012.
>
> K. He, X. Zhang, S. Ren, and J. Sun. Deep residual learning for image recognition. In Proceedings of the IEEE conference on computer vision and pattern recognition, pages 770–778, 2016.
>
> > 8, line 297, how is D obtained in the analysis? is it dependent on μ or L?
>
> The constant $D$ does not appear in the analysis. We make this assumption on line 297 only in order to illustrate the interesting connection with convex case. In fact such additional assumption is not restrictive since it holds with high probability due to convergence of the method (and compactness of the set of solutions $X^*$). In this sense, $D$ is independent of $\mu$ and $L$, but only depends on the distance from the initial point to the set $X^*$.
>
> > 9, in theorem 3, why is the step size ηt dependent on n? when n is large, the learning rate will be extremely small, right?
>
> Yes, this is correct and it is expected in a finite sum setting. To the best of our knowledge, it is the case for all variance reduction methods in the finite sum case. The main challenge in the finite sum case is usually to improve the dependence on $n$, while keeping the dependence on $\varepsilon_f$ the same as for deterministic methods.
>
> > 10, why did not include some numerical results?
>
> We kindly refer the reviewer to our answers above. We provide numerical simulations in the revision. The experimental results are well-aligned with our theoretical study.

---

> > ### Comment · Reviewer_FAsV · 2022-08-09
> > **Further questions**
> >
> > Many thanks for addressing my comments. But there are still some concerns left.
> >
> > 1, How is ensured that D is bounded without any projection? is it proved in this paper?
> >
> > 2, numerical experiments on the RL problem for testing PAGER is not quite reasonable except for the tabular case as this VR algorithm still needs the full batch. In this case, STORM type of algorithms is preferred.

---

> > > ### Author Response · Authors · 2022-08-09
> > > **Further clarifications**
> > >
> > > We thank the reviewer for reading our responses and the follow-up questions.
> > >
> > > 1. Note that our analysis on SGD and PAGER under the global KL condition throughout the paper does not require boundedness on the iterates. This side remark we had on the potential connection to stochastic convex optimization says that assuming  iterates are further bounded (e.g., by imposing some regularity assumption such cooercivity or projection), then our result can apply to stochastic convex problems and leads to the "seemingly first" last iterate convergence with PAGER, which could be of independent interest.
> > >
> > >
> > > 2. There exist many VR variants in the literature, among which PAGE and STORM are the few loopless algorithms. The reviewer is correct that STORM does not require mini-batch unlike PAGE. We choose PAGE as our workhorse over STORM due to several considerations:
> > >
> > > - On the theory side, PAGE achieves the best known complexity for the special 2-PL case [Li et al, 2021], whereas other popular VR methods such as SVRG have worse dependence on $n$ for 2-PL objectives [Reddi et al, 2016] and whether STORM can achieve the optimal complexity for 2-PL or even strongly convex objectives remains elusive, to the best of our knowledge.
> > > In this regard, PAGE looks more promising.
> > >
> > > - On the practice side, especially for RL applications, taking mini-batching is not an issue with on-policy sampling. Recent ICML work (https://arxiv.org/pdf/2202.00308.pdf) also provided numerical evidence showing that PAGE sometimes outperforms STORM in RL tasks (see Figure 2 in their paper).

---

> ### Author Response · Authors · 2022-08-08
> **Author-Reviewer discussion period**
>
> Dear Reviewer FAsV, as the discussion period is ending today, we appreciate if the reviewer could take a chance to read our responses and let us know if we have addressed your questions.  Do you have any further questions that we can clarify?

---

### Author Response · Authors · 2022-08-02
**General Response to All Reviewers**

We thank the reviewers for their feedback and insightful comments. We are glad that the reviewers appreciate that the studied problem is **well motivated** (Reviewer FAsV, cXuP, ws9Y), the work is **“well-structured and easy to understand”** (reviewer ekoT) and “**generalizing the proofs is not straightforward from the prior work**” (reviewer ws9Y). At the same time, we took all the comments seriously and will soon upload detailed responses to each comment. We also made the necessary clarifications in the revised version of the paper. We would be happy to provide the responses to any further questions and comments of the reviewers!

A list of key changes (highlighted in blue in the revision):

$\bullet$ (Section 3.2) We update this subsection and clarify the result about the sample complexity of SGD. In particular, we show that under Assumption 4, increasing the cost of estimator $b_k$ does not improve the total sample complexity.

$\bullet$ (Appendix A) We expand our section with examples of KL functions. In particular, we provide an example of KL function, which is not $\alpha$-PL.

$\bullet$ (Appendix C) We extend the discussion about the challenges of analyzing variance reduction for KL functions.

$\bullet$(Appendix E) We add numerical experiments on the reinforcement learning (RL) policy optimization problem, which satisfies 1-PL assumption.

We emphasize that the main contributions of our work are convergence analysis of SGD for non-convex functions under the global KL condition and, in particular, for the important class of functions with $\alpha$-PL condition. More importantly, we introduced a variance reduction algorithm (PAGER) that achieves $O(\epsilon^{-2})$ sample complexity for an important class of 1-PL functions that appears in RL problems. This is the first optimal algorithm that matches the lower bound known for stochastic convex optimization.

---

### Author Response · Authors · 2022-08-07
**Feedback request**

We thank the reviewers once again for their reviews. In our responses, we addressed all concerns of the reviewers.

Considering that the discussion period ends in two days, we kindly ask the reviewers to let us know whether our replies are convincing and whether additional clarifications are required. Furthermore, we would be happy to address new questions and criticism in case of any.

We thank the reviewers in advance.

---

### Meta-Review · Area_Chair_xP1a · 2022-08-25

**Recommendation:** Accept
**Confidence:** Less certain

**Metareview:**

This paper offers an analysis for SGD, and then a variance reduced method PAGER, under the general KL assumption. This is a very large family of functions, that includes many interesting non-convex objectives, and thus is interesting for the community. And quoting one of the reviewers "The rates provided by the paper (in Corr. 1 and Thm. 3) recover the best prior rates when specialised to the setting of (\alpha = 2), and extend these settings for general (\alpha) and for SGD, under assumption 4 specifically. This means we gain more generality at no additional cost." The reviewers also eventually agreed that the technical novelties introduced to establish the proof are also interesting and new.

**Award:**

No

---

### Decision · Program_Chairs · 2022-09-14

Accept